# Dynamic mechanism of an extremely severe saltwater intrusion in the Changjiang Estuary in February 2014

Jianrong Zhu[1], Xinyue Cheng[1], Linjiang Li[1], Hui Wu[1], Jinghua Gu[1], Hanghang Lyu[1]

[1]State Key Laboratory of Estuarine and Coastal Research, East China Normal University, Shanghai, 200241, China

*Correspondence to*: Jianrong Zhu (jrzhu@sklec.ecnu.edu.cn)

**Abstract.** Estuarine saltwater intrusions are mainly controlled by river discharge and tides. Unexpectedly, an extremely severe saltwater intrusion event occurred in February 2014 in the Changjiang Estuary under normal river discharge conditions. This intrusion cut off the freshwater input for 23 days into the Qingcaosha Reservoir, which is the largest estuarine reservoir in the

world, creating a severe threat to water safety in Shanghai. No similar catastrophic saltwater intrusion has occurred since records of salinity in the estuary have been kept. During the event, a persistent and strong northerly wind existed, with a maximum speed of 17.6 m s[-1], lasting 9 days and coinciding with a distinct water level rise. Our study demonstrates that the extremely severe saltwater intrusion was caused by this northerly wind, which drove substantial landward net water transport to form a horizontal estuarine circulation that flowed into the North Channel and out of the South Channel. This landward net

water transport overpowered the seaward-flowing river runoff and transported a large volume of highly saline water into the North Channel. The mechanisms of this severe saltwater intrusion event, including the northerly wind, residual water level rise, landward water transport and resulting horizontal circulation, etc., were systematically investigated.

## 1 Introduction

Saltwater intrusion is a common phenomenon in estuaries where freshwater and saltwater converge and is mainly

controlled by tides and river discharge (Prandle, 1985; Simpson et al., 1990; Geyer, 1993), but it can also be affected by wind stress (Chen and Sanford, 2009; Aristizảbal and Chant, 2015; Duran-Matute et al., 2016; Giddings and Maccready, 2017) and vertical mixing (Simpson and Hunter, 1974; Prandle and Lane, 2015). Along-estuary winds can strain density gradients, and the associated destruction or enhancement of stratification depending on wind direction, and the entrainment depth ratio (Chen & Sanford, 2009). Wind-driven sea-level setups at the mouth of estuaries can produce landward flows that outcompete river

runoff, resulting in the net, landward advection of salt (Aristizabal & Chant, 2015). For multi-inlet coastal systems, residual (horizontal) circulation is influenced by winds, which alters salt transports in each inlet (Duran-Matute et al., 2016). The upwelling and downwelling favorable wind can significantly influence the estuarine exchange flow (Giddings, and MacCready, 2017). Saltwater intrusions can produce estuarine circulation (Pritchard, 1956) and affect stratification (Simpson et al., 1990), thereby influencing sediment transport, producing peak estuarine turbidities (Geyer, 1993), and degrading the freshwater

quality (Zhu et al, 2013). Therefore, the study of estuarine saltwater intrusions has scientific relevance for understanding circulation, sediment, and ecological dynamics and can facilitate the effective utilization of estuarine freshwater resources.

Changjiang, also known as the Yangtze River, is one of the largest rivers in the world and discharges large amounts of freshwater ($9.24 \times 10^{11}$ m$^3$) into the East China Sea each year (Shen et al., 2003), with seasonal variations in river discharge ranging from a maximum monthly mean of 49,850 m$^3$ s$^{-1}$ in July to a minimum of 11,180 m$^3$ s$^{-1}$ in January (Zhu et al., 2015).

The Changjiang Estuary is characterized by multiple bifurcations (Fig. 1). The tides in the Changjiang Estuary are semidiurnal, have biweekly spring-neap signals and are highly energetic sources of water movement. The maximum tidal range reaches 3.38 m and the minimum tidal range reaches 0.64 m at the Baozhen hydrological station (Zhu et al., 2015). The maximum tidal current amplitude reaches approximately 2.0 m s$^{-1}$ at the river mouth during the spring tide. The prevailing monsoon climate results in a strong northerly wind of 5.5 m s$^{-1}$ during winter and a southeasterly wind of 5.0 m s$^{-1}$ during summer (Zhu

et al., 2015). Saltwater intrusion in the Changjiang Estuary is also mainly determined by river discharge and tides (Song and Mao, 2002; Gu et al, 2003; Shen et al., 2003; Luo and Chen, 2005; Qiu et al. 2012; Chen et al, 2019a) but is also influenced by wind (Xue et al., 2009; Wu et al., 2010; Li et al., 2012; Ding et al., 2017; Zhang et al., 2019) and topography (Li et al., 2014; Chen et al., 2019b). The impact of wind on saltwater intrusion has been studied, but only with a climatic wind (Xue et al., 2009; Qiu et al. 2012; Chen et al, 2019a ), and a strong northerly wind induced by ordinary cold fronts in winter lasting 1-

2 days, which could cause a change in the observed salinity (Li et al., 2012). Xue et al. (2009) pointed out that a northerly wind tends to enhance the saltwater intrusion in the North Branch by reducing the seaward surface elevation gradient forcing. Wu et al. (2010) and Li et al. (2012) simulated the pure wind-driven current that flows into the North Channel and out of the South Channel with climatic wind to explain that the northerly wind can enhance saltwater intrusion in the North Channel, and weaken it in the South Channel. Zhang et al. (2019) reported that the frequency of saltwater intrusion events in the Changjiang

Estuary is increasing in recent years due to increasing frequency of winter storms passing East China Sea.

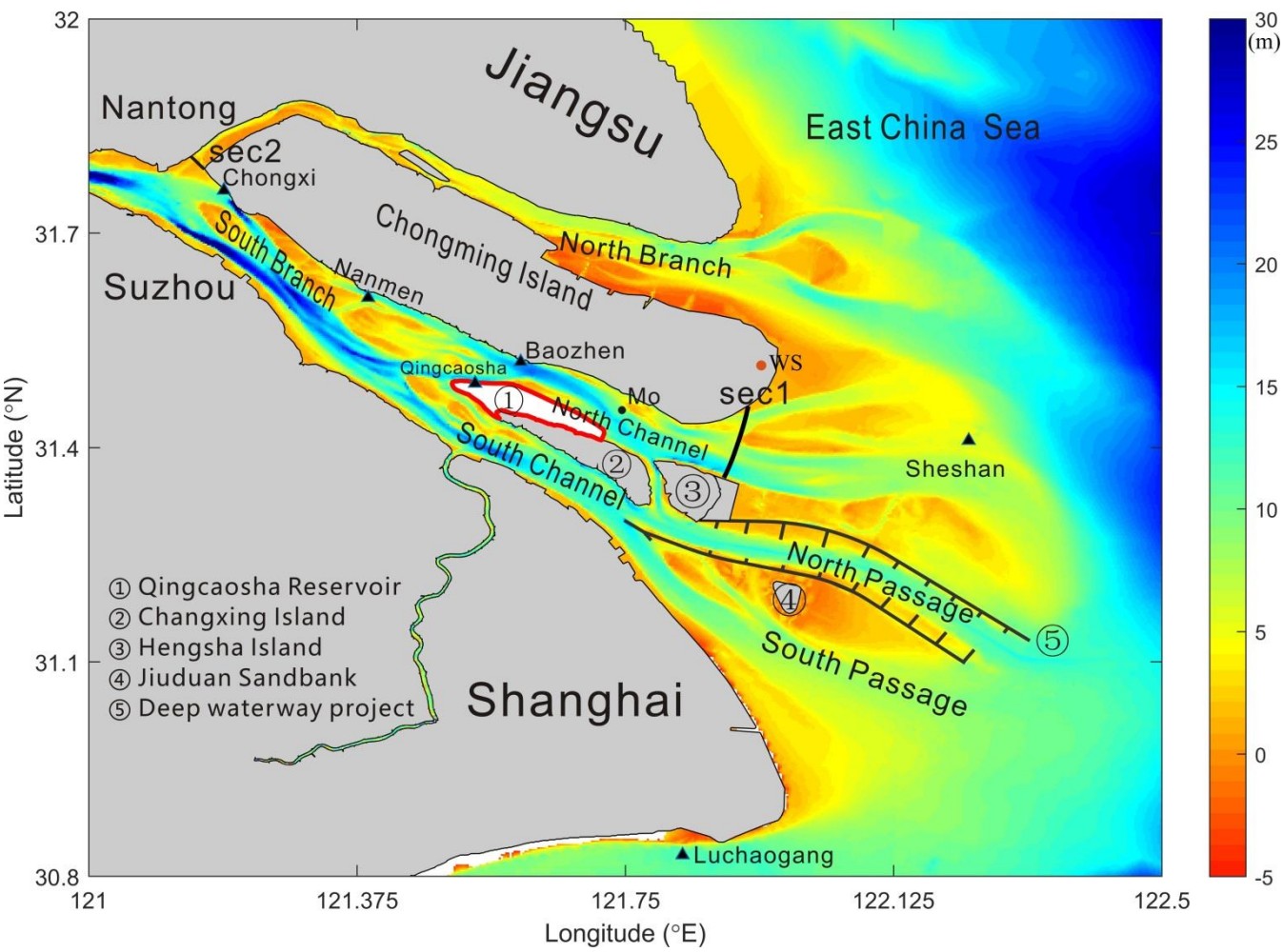

**Figure 1: Topography of the Changjiang Estuary. The black triangles indicate the locations of hydrologic stations Chongxi, Nanmen, Baozhen, Qingcaosha, Sheshan and Luchaogang. WS is the location of the weather station at the Chongming eastern shoal. sec1 and sec2 are transects at the river mouth of the North Channel and upper reaches of the North Branch, respectively, and Mo is the model output site for terms in the momentum equations.**

The estuaries of large rivers are often associated with growing populations and economic activity, leading to complex challenges in managing environmental conditions. Freshwater supplementation is of vital importance for industrial and home use, but estuarine freshwater is subject to frequent sea water intrusions. The Changjiang Estuary, which is surrounded by fast-developing cities, such as Shanghai, Suzhou and Nantong city, is threatened by saltwater intrusions in the winter season. Numerous efforts have been made in recent decades to meet the astronomic freshwater demand of the megacity of Shanghai, which has a population of over 24 million people. The water resources in Shanghai were strategically transferred from the Huangpu River to the Changjiang Estuary in 2010, when the largest estuarine reservoir in the world, the Qingcaosha Reservoir, was built (shown in Fig. 1). This reservoir has an effective capacity of $4.35 \times 10^8$ m$^3$ and provides a daily water supply of $7.19 \times 10^6$ m$^3$ for the 13 million people in the main districts of Shanghai, accounting for 70% of the total freshwater in the city.

However, the Qingcaosha Reservoir is frequently influenced by saltwater intrusions, particularly during the dry season. Limited by reservoir capacity, long-lasting saltwater intrusions are extremely harmful (Chen et al, 2019a). The most characteristic type of saltwater intrusion in the estuary is the saltwater spillover (SSO) from the North Branch into the South Branch (Shen et al., 2003; Wu et al., 2006; Lyu and Zhu, 2018). The shallow and funnel-shaped topography helps to prevent runoff from entering the North Branch, especially during the dry season, and produces a greater tidal range in the North Branch

than in the South Branch. The saltwater from the SSO is transported downstream by runoff and arrives in the middle reaches of the South Branch during the subsequent neap tide, threatening the water supplies of reservoirs located in the estuary.

An extremely severe saltwater intrusion event occurred in February 2014 in the Changjiang Estuary under normal river discharge conditions and seriously influenced the water intake of the Qingcaosha Reservoir and threatened water safety in Shanghai. Two severe saltwater intrusion events were recorded in the Changjiang Estuary in the dry seasons of 1979 and 1999,

which were caused by very low river discharge with a mean value of 7485 $m^3 s^{-1}$ from January 1 to March 15, 1979, and 9246 $m^3 s^{-1}$ from January 1 to March 16, 1999, both lasting 2.5 months (Zhu et al., 2013). However, the mean monthly river discharge in February 2014 was 11,510 $m^3 s^{-1}$, which approached the annual climatic mean value of 12,430 $m^3 s^{-1}$ in February from 1950 to 2019. No similar catastrophic saltwater intrusion event has occurred since salinity data have been recorded, even when there was a much lower river discharge in the estuary, such as in the dry seasons of 1979 and 1999. Because the river discharge in

February 2014 was close to the monthly mean discharge since 1950, river discharge was not the cause of the extremely severe saltwater intrusion event. The tides in February in different years have been similar, as has the estuarine topography over the last ten years. What was the reason for the extremely severe saltwater intrusion event in the Changjiang Estuary in February 2014? In this paper, the dynamic mechanism of the event was studied using observation data and a numerical model.

## 2 Methods

### 2.1 Numerical model

The numerical model used in this study was based on ECOM-si (Blumberg, 1994) and later improved for better studying hydrodynamics and substance transport (Chen et al., 2000; Wu and Zhu, 2010). The model uses a sigma coordinate system in the vertical direction and a curvilinear nonorthogonal grid in the horizontal direction (Chen et al., 2004). The model domain for saltwater intrusion covers the Changjiang Estuary and its adjacent sea region (Fig. 2a). Datong hydrological station is on

the western open boundary of the Changjiang River.

The model simulations in this study covered the period from January 1 to February 28, 2014. The daily river discharge recorded at the Datong hydrologic station was used in the model as the river boundary condition (Fig. 2b). At the open sea boundaries of the model, the momentum open boundaries were driven by total water levels, which are composed of the residual water level and tidal level:

$$\zeta = \bar{\zeta} + \sum_i^{16} a_i \cos(\omega_i + g_i) \qquad (1)$$

where $\zeta$ is the total water level and $\bar{\zeta}$ is the residual (mean) water level reflecting the shelf current. The tidal level is calculated by combining the 16 main tidal constituents ($M_2$, $S_2$, $N_2$, $K_2$, $K_1$, $O_1$, $P_1$, $Q_1$, $MU_2$, $NU_2$, $T_2$, $L_2$, $2N_2$, $J_1$, $M_1$, and $OO_1$) with harmonic constants, $a$ and $g$, which are the amplitude and phase of the tidal constituent, respectively, derived from the NaoTide dataset (NaoTide, 2004), and $\omega$ is the frequency. The method for determining the mean water level in Eq. (1) is a critical issue for correctly simulating saltwater intrusion in the Changjiang Estuary under a strong northerly wind. In this study, the mean water level was simulated by a large domain model encompassing the Bohai Sea, Yellow Sea and East China Sea (Fig. 2b, the model grids and domain), which is driven by ocean circulation, tide, river discharge and sea surface wind to simulate the water level, current and salinity (Wu et al., 2011).

## 2.2 Numerical validation

The numerical model has been extensively calibrated, validated and applied in a number of previous studies of the Changjiang Estuary, which have shown that the model can reproduce the observed water level, current and salinity with high simulation accuracy (Wu and Zhu, 2010; Qiu and Zhu, 2015; Lyu and Zhu, 2018). Detailed descriptions of the model validation process can be found in the literature mentioned above. In this study, the model was validated with the observed water level at Baozhen, Shenshan and Luchaogang stations and salinity at Nanmen, Baozhen, Chongxi and Qingcaosha stations during the extremely severe saltwater intrusion event, and the results showed that modeled salinities were well consistent with the observed values (shown in Fig. 3 and Fig. 6 and analyzed in Sect. 3.2).

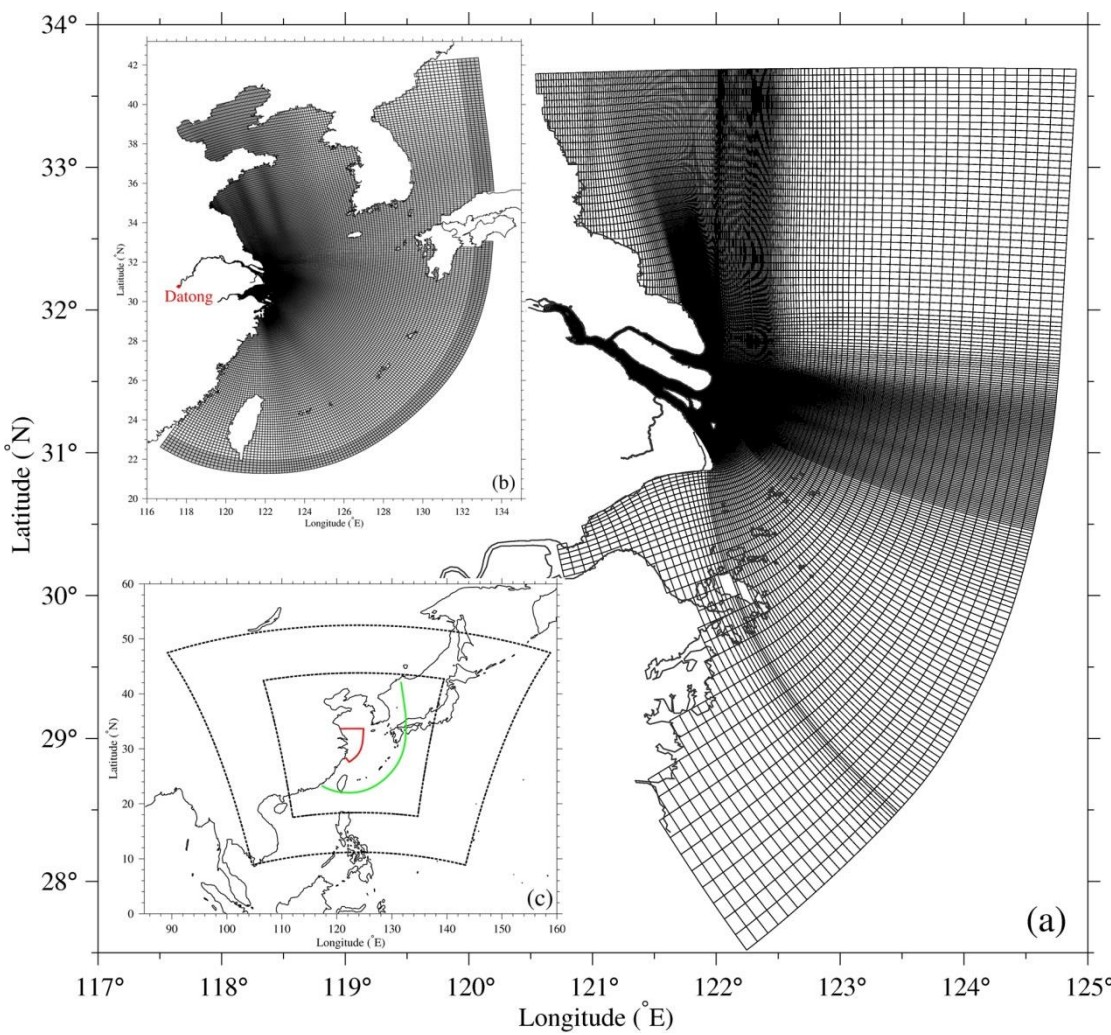

**Figure 2: Model grids of the Changjiang Estuary (a) and model grids of the Bohai Sea, Yellow Sea and East China Sea (b). Domains of the models (c); within the red line: the Changjiang Estuary model domain; within the green line: model domain of the Bohai Sea, Yellow Sea and East China Sea; black dashed lines: the two-fold nested WRF model domain.**

## 3 Results

### 3.1 Observed data

The observed data of salinity, water level and wind were used to study the extremely severe saltwater intrusion event in February 2014. The observed salinity and wind data were from the State Key Laboratory of Estuarine and Coastal Research, East China Normal University, and the observed water level data were from the Shanghai Hydrology Administration. At the Baozhen hydrologic station, the salinity was normal before February 8, 2014 (in relation to January and February in other years) but rose to a very high value with a peak of 20.1 from February 9 to 20, which was beyond expectations and had never

occurred (Fig. 3a). At Nanmen station, the salinity was also abnormally high, with a peak salinity of 12.4. At Chongxi station, the salinity was high, with a peak salinity of 6.5 induced by the SSO. At Qingcaosha station, the salinity was greater than 0.45

(the salinity standard for drinking water, similarly hereinafter) from February 4 to 26, with a peak salinity of 8.6. Thus, the continuous period of unsuitable drinking water reached 23 days, causing a serious threat to the water intake of the Qingcaosha Reservoir and water safety in Shanghai. The observed salinities at hydrologic stations indicated that there was an extremely severe saltwater intrusion event in February 2014.

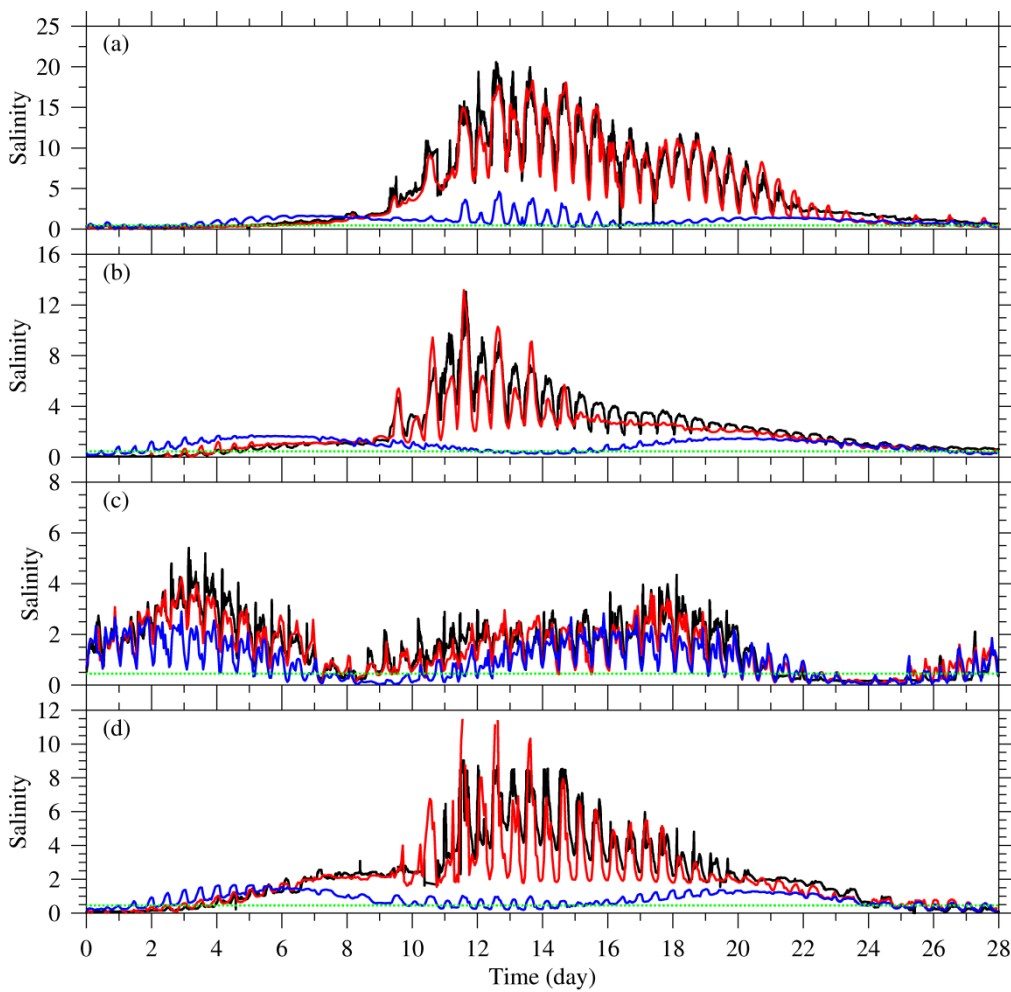

**Figure 3: Temporal variations in salinity in February 2014 at hydrologic stations. a: Baozhen station; b: Nanmen station; c: Chongxi station; c: Qingcaosha station. Black line: measured salinity; blue line: simulated salinity under climatic wind and residual water level conditions at open sea boundaries (Exp 1); red line: simulated salinity under a realistic wind and residual water levels at the open sea boundaries (Exp 2). The dashed green line represents salinity of 0.45, which is the standard for drinking water.**

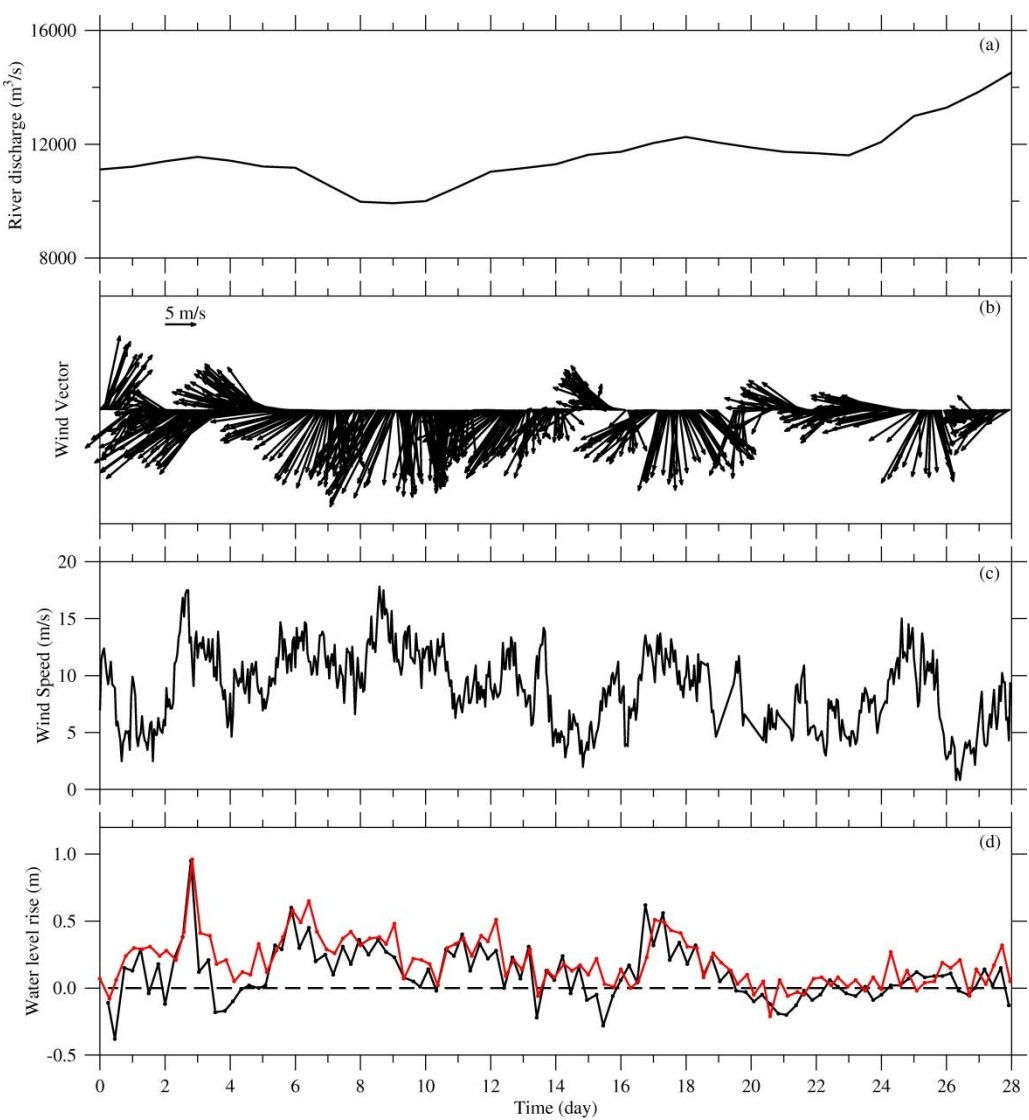


**Figure 4: Temporal variations in the measured river discharge at Datong station (a), wind vector (b) and wind speed (c) at WS, and water level rise obtained by subtracting the data in the tide table from the measured water level at Sheshan station (black line) and Luchaogang station (red line) (d) in February 2014.**

The mean monthly river discharge recorded at Datong hydrologic station (Changjiang Water Resources Commission) in

February 2014 was 11,510 m³/s (Fig. 4a), which approached the annual climatic mean value of 12,430 m³/s in February from 1950 to 2019. Thus, river discharge was not the cause of the extremely severe saltwater intrusion event.

The weather station at the Chongming eastern shoal (location is shown in Fig. 1, WS) recorded a persistent and strong northerly wind (2 minutes average) from February 6 to 14, 2014, lasting nine days with an average wind speed of 10.01 m/s and a maximum wind speed of 17.6 m/s on February 10 (Fig. 4b, 4c). After a two-day southerly wind existed from February

15 to 16, there was a four-day strong northerly wind from February 17 to 20, 2014. At the same time, the Sheshan and

Luchaogang hydrologic stations recorded a distinct water level rise, with a peak value of more than 0.5 m during the strong northerly wind (Fig. 4d). The water level rise couldn't be directly observed, and was obtained by subtracting the data in the tide table from the measured water level value.

The observed data indicated that an extremely severe saltwater intrusion in the Changjiang Estuary occurred in February 2014; meanwhile, the river discharge was normal, and a persistent and strong northerly wind was present.

## 3.2 Modeled results

To reveal which dynamic factor caused the extremely severe saltwater intrusion event in February 2014, two numerical experiments were designed. The first experiment considered climatic wind conditions at the sea surface and residual water levels at open sea boundaries (Exp 1), and the second considered a realistic wind in February 2014 and residual water levels at the open sea boundaries (Exp 2). The river discharge at the upper boundary at Datong station and tide at the open sea boundary were the same in Exp 1 and Exp 2.

In Exp 1, the monthly mean wind field dataset with a spatial resolution of 0.25 °×0.25 ° and a temporal resolution of 6 h from the National Centers for Environmental Prediction/Quick Scatterometer (NCEP/QSCAT, 2014) was used. The climatic wind field in February is a northerly wind with a speed of approximately 7 m s$^{-1}$ over the Yellow Sea and East China Sea (Fig. 5a). The residual water level and residual current were simulated by the model, including the Bohai Sea, Yellow Sea and East China Sea. Forced by the northerly wind, the residual water level was 10~20 cm away from the Changjiang River mouth and was mainly induced by river discharge and a climatic northerly wind, and the residual surface current flows southward off the Changjiang mouth (Fig. 5b). The residual water level was interpolated into the open sea boundary of the saltwater intrusion model as an external forcing condition.

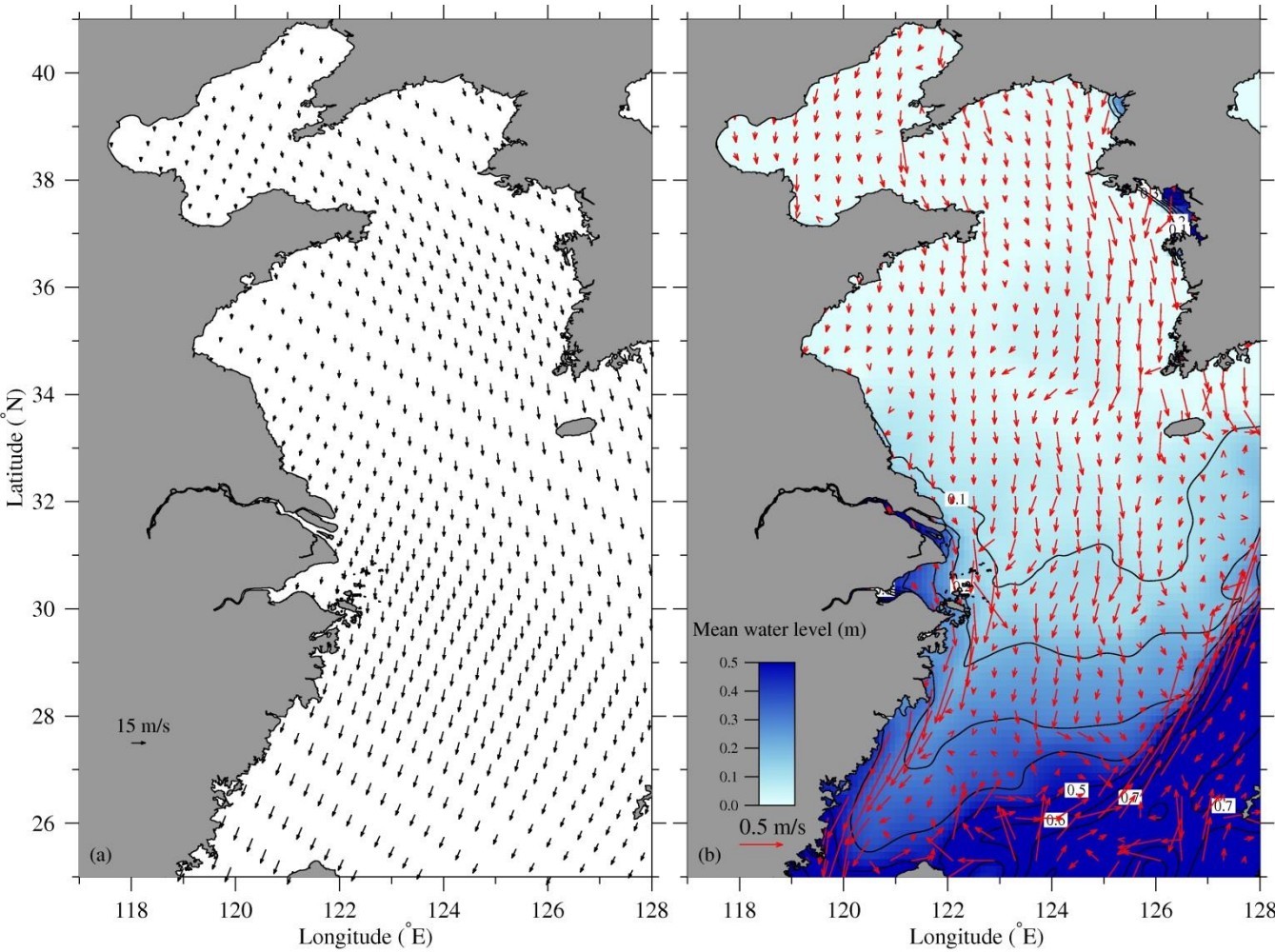


**Figure 5: Distributions of the climatic wind field (a) and residual (mean) water level and surface current (b) in February.**

The observed and modeled water level at Baozhen, Sheshan and Luchaogang stations is fairly consistent with the observed water level (Fig. 6), meaning that the water level is mainly determined by tide and river discharge inside the river mouth. However, during the neap tide from February 6 to 9, 2014, the modeled water level was approximately 10-30 cm lower than
the observed level because the wind was not a realistically strong northerly wind.

The modeled salinity at Baozhen, Nanmen, Chongxi and Qingcaosha stations is shown in Fig. 3. Under climatic wind, the modeled salinity was significantly lower than the observed salinity at Baozhen, Nanmen, and Qingcaosha stations and was distinctly lower at Chongxi station, which is similar to the normal saltwater intrusion regularly occurring under monthly mean river discharge and wind in the dry season (Shen et al., 2003; Zhu et al., 2015).

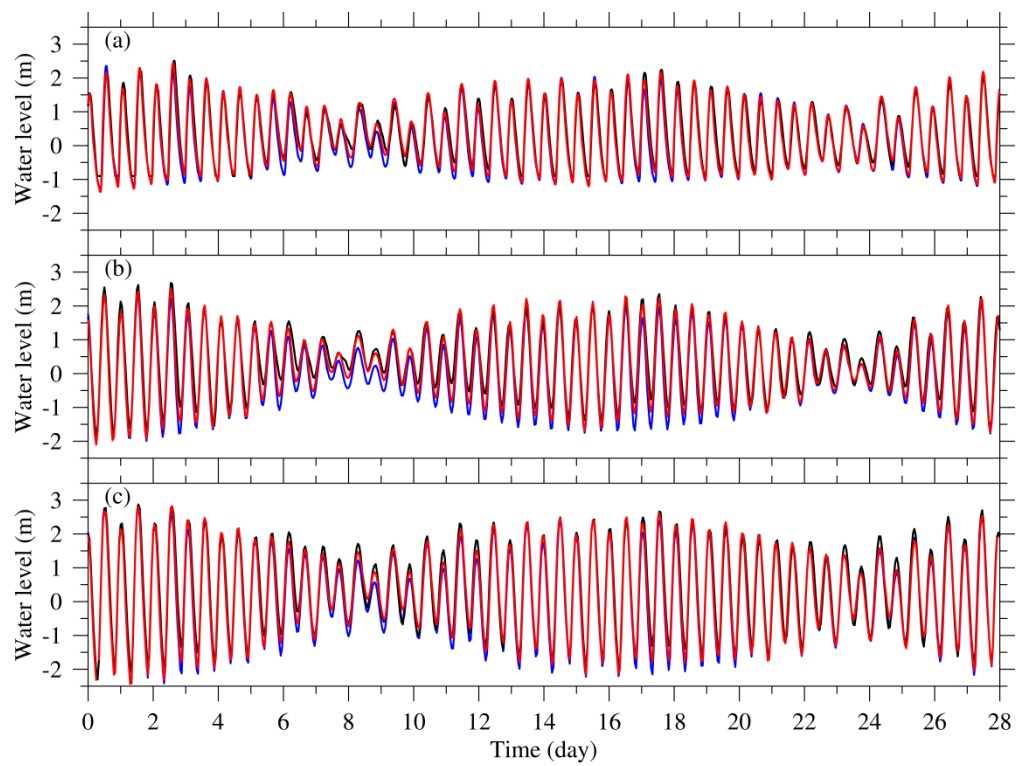


**Figure 6: Temporal variations in water level in February 2014 at hydrologic stations. a: Baozhen station; b: Sheshan station; c: Luchaogang station. Black line: measured data; blue line: simulated data in Exp 1; red line: simulated data in Exp 2.**

The residual unit width water flux at the surface from February 10 to 13, 2014, flowed seaward into the South Branch, North Channel, North Passage and South Passage but flowed landward into the North Branch due to its funnel shape and tidal
Stokes transport (Qiu and Zhu, 2015), which means that the South Branch, North Channel and South Channel are the main channels for discharging river water into the sea (Fig. 7a). The distribution of time-averaged surface salinity indicated that there was a strong salinity front near the river mouth, which was caused by the confluence and mixing of fresh river water with sea water (Fig. 7b). The North Branch was occupied by highly saline water due to its topography. The funnel shape amplifies the tide in its upper reaches, and the wider tidal flat in the upper reaches lowers the amount of river discharge inflow (Wu et
al., 2006; Lyu and Zhu, 2018). Salinity in the South Branch was less than 0.45, meaning that there was a wide area of fresh water near the water intake of the Qingcaosha Reservoir. Among the South Passage, North Passage and North Channel, saltwater intrusion was the strongest in the South Passage and weakest in the North Channel, which is consistent with previous studies (Li et al., 2014; Wu et al., 2010; Lyu and Zhu, 2018).

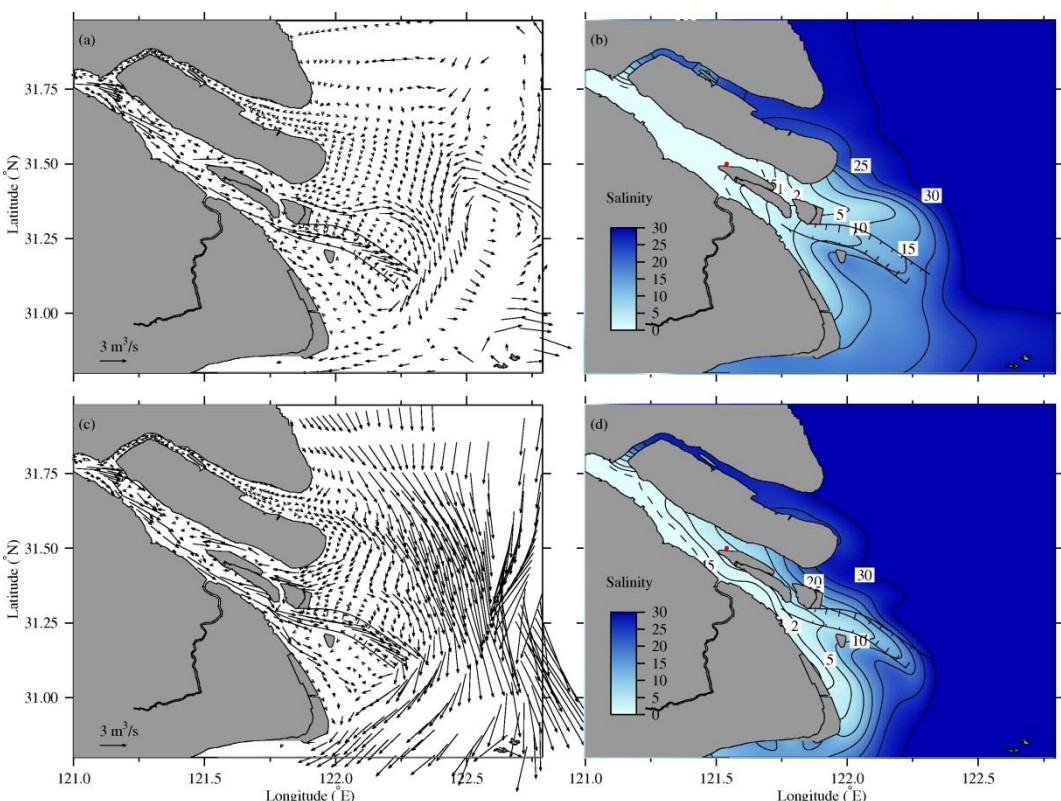

**Figure 7: Distributions of residual unit width water flux (a, c) and time-averaged surface salinity (b, d) from February 10 to 13, 2014, from Exp 1 (upper panel) and Exp 2 (lower panel). The dashed isohaline represents a salinity of 0.45; the red dot denotes the location of water intake of Qingcaosha Reservoir (similarly hereinafter).**

The residual water flux across the transverse section in the North Channel (sec 1 labeled in Fig. 1) from February 6 to 24, 2014, flowed seaward due to runoff force and decreased during the neap tide from February 9 to 11 (Fig. 8a). For the bifurcated Changjiang Estuary, more river discharge flowed into the North Channel during spring tide and into the South Channel during neap tide (Li et al., 2010; Lyu and Zhu, 2018). The residual salt flux across this section flowed landward from February 10 to 13, 2014, during the later neap tide and the subsequent early-middle tide due to weaker tidal mixing and the saltwater wedge and flowed seaward during other tidal conditions (Fig. 8b).

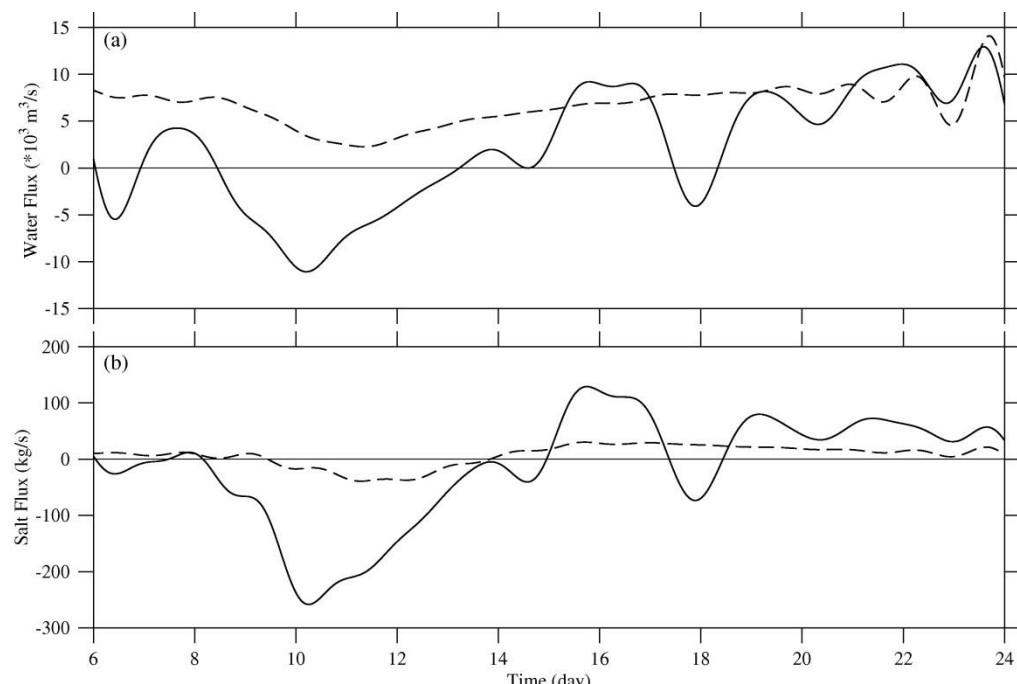

Figure 8: Temporal variations in residual water flux (a) and salt flux (c) across Sec 1 in the North Channel from February 6 to 24, 2014. Dashed line: Exp 1; solid line: Exp 2. A positive value represents seaward flux, and a negative value represents landward flux.

In Exp 1, the extremely severe saltwater intrusion event in February 2014 was not replicated, whereas the saltwater intrusion was simulated with Exp 2. The realistic wind was simulated by a mesoscale atmospheric model: the Weather Research Forecasting (WRF) model with two nested domains (Fig. 2c). The NCEP reanalysis dataset was used to establish the initial and boundary conditions of the WRF model. A comparison between the WRF modeled wind vector and the measured wind vector at the weather station located at the Chongming eastern shoal showed that the persistent and strong northerly wind in February 2014 was well simulated by the WRF model (figure omitted).

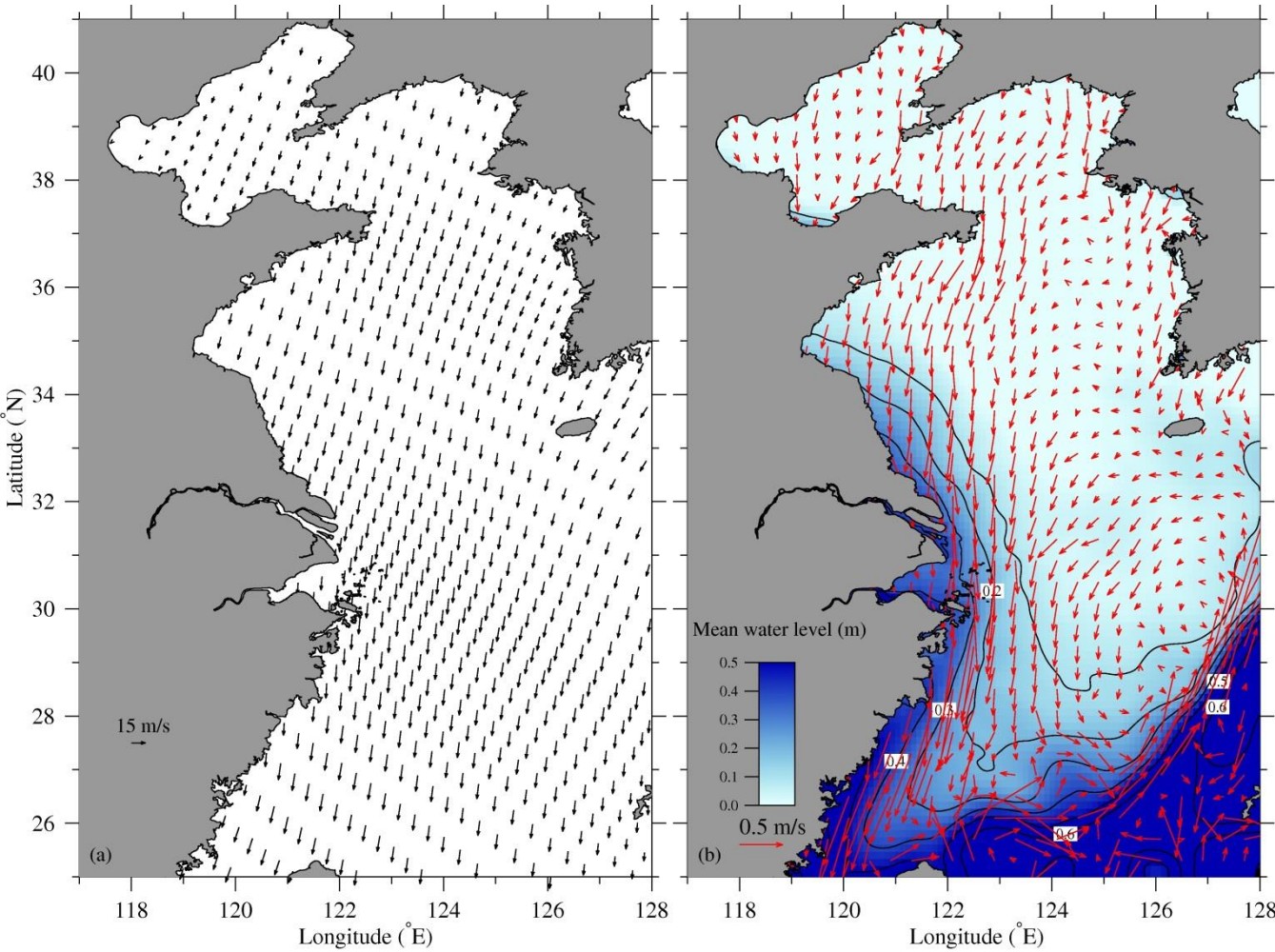

**Figure 9: Distributions of the temporally averaged wind field from February 7 to 14, 2014, as simulated by the WRF model (a), and the time-averaged (mean) water level and surface current from February 10 to 13, 2014, as simulated by the model encompassing the Bohai Sea, Yellow Sea and East China Sea (b).**

The simulated temporally averaged wind field over the Yellow Sea and East China Sea from February 7 to 14, 2014, indicated that the northerly wind speed reached approximately 18 m s$^{-1}$ (Fig. 9a), which was much higher than that observed at the weather station at the Chongming eastern shoal (Fig. 4b, c). The residual (mean) water level simulated by the model of the Bohai Sea, Yellow Sea and East China Sea showed that the strong northerly wind induced a significant high water level along the coast of China during the extremely severe saltwater intrusion event from February 10 to 13, 2014 (Fig. 9b). The residual water level was interpolated into the open sea boundaries of the saltwater intrusion model. The temporal variations in wind vector and wind speed (b) at Sheshan island (labeled in Fig. 1) in February 2014 simulated by WRF (Fig. 10) was similar with the one observed at Chongming eastern shoal (Fig. 4b, c), indicating again that there existed a persistent and strong northerly wind in February 2014.

In comparison to the results of Exp 1, the simulated water level in Exp 2 at Baozhen, Sheshan and Luchaogang stations was closer to the observed water level, especially during the neap tide from February 7 to 11, 2014 (Fig. 6). The difference between the observed water level and modelled water level under climatic wind can be considered the water level rise by the northerly strong wind. It is seen from the Fig. 6 that the water level rise at low water level from February 5 to 12, 2014 was approximately 0.35, 0.40, and 0.30 m at Baizhen, Sheshan and Luchaogang station, respectively. The simulated salinity at Baozhen, Nanmen, Chongxi and Qingcaosha stations was significantly improved and was very consistent with the observed salinity in terms of both magnitude and phase (Fig. 3). The salinity modeled under climatic wind and residual water level conditions at open sea boundaries was significantly lower than the observed salinity, and the salinity variation process was successfully replicated under realistic wind and residual water levels at the open sea boundaries. Therefore, the persistent and strong northerly wind played an important role in the extremely severe saltwater intrusion event.

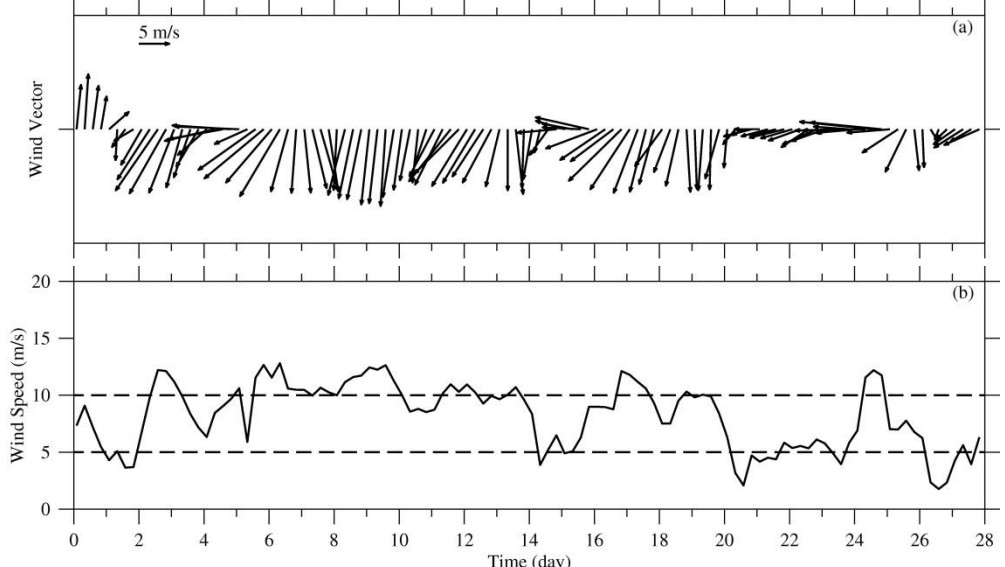

**Figure 10: Temporal variations in wind vector (a) and speed (b) in February 2014 simulated by WRF.**

The temporal variations in the observed and modeled water levels at Baozhen, Sheshan and Luchaogang stations (Fig. 6) and salinity at Chongxi, Nanmen and Qingcaosha hydrologic stations in February 2014 (Fig. 3) show that the model can reproduce the salinity variation processes involved in the extremely saltwater intrusion event. The model was validated again, and the results are quite robust.

The long-lasting persistent and strong northerly wind produced a strong southward current with a speed of 40-60 cm s$^{-1}$ and landward Ekman water transport under the Coriolis force, resulting in a high water level of more than 30 cm along the coast of China (Fig. 9b), which was confirmed by the observed water level rise at the Sheshan and Luchaogang stations (Fig. 4d). Because the North Branch is very shallow, the landward Ekman water transport was weaker, flowing along the north side and flowing out along the south side only near the river mouth (Fig. 7c). However, the North Channel is deeper and wider and

located on the north side of the South Branch, which is conducive to strong landward Ekman water transport in the North Channel.

The residual unit width water flux from February 10 to 13, 2014, flowed landward into the North Channel, meaning that the landward Ekman transport induced by the persistent and strong northerly wind surpassed the seaward runoff (Fig. 7c). The Changjiang is a large river, and its river discharge was 11,510 $m^3$ $s^{-1}$ in February 2014, so it is unexpected that wind-driven landward water transport could surpass the strong seaward runoff. The current flowed into the North Channel and flowed out of the South Channel, forming a horizontal circulation and leading to highly saline sea water into the North Channel.

The residual water and salt flux across the transverse section in the North Channel from February 9 to 13, 2014, flowed landward with a maximum value of 12,000 $m^3$/s and 360 kg/s, respectively (Fig. 8), which was consistent with the distribution of residual unit width water flux. In Exp 1, the residual water flux was seaward due to runoff, and the landward residual salt flux from February 10 to 12, 2014, was very small compared to that in Exp 2. Therefore, the extremely severe saltwater intrusion event in February 2014 was caused by the persistent and strong northerly wind, which produced a strong landward

net transport and abnormal water level rise, induced a horizontal circulation and brought highly saline water into the North Channel.

## 4 Discussion

This section systematically discusses the potential mechanism of the extremely severe saltwater intrusion with regard to the SSO, how the wind affects the individual terms in the momentum equations, the relationship between residual water level and wind, the generation of Ekman transport and the resulting horizontal circulation, and the duration of the severe saltwater

intrusion induced by the northerly wind.

### 4.1 What was the contribution of the SSO to the saltwater intrusion event?

The most obvious feature of saltwater intrusion in the Changjiang Estuary is the SSO. Previous studies showed that the SSO is the main source of saltwater intrusion in the upper and middle reaches of the South Branch and the main saltwater

source of the reservoirs (Shen et al. 2003; Wu et al. 2006; Zhu et al., 2013; Lyu and Zhu, 2018). What was the extent of the contribution of the SSO to the extremely severe saltwater intrusion in February 2014? A transect Sec 2 at the upper reaches of the North Branch (location labeled in Fig. 1) was set to calculate water and salt flux.

In Exp 1, the water flux from February 6 to 8, 2014, and salt flux on February 7, 2014, were transported from the South Branch into the North Branch (Fig. 11). This phenomenon occurred during a neap tide (indicated by the water level in Fig. 6),

while in the other tidal patterns, the water and salt flux was transported from the North Branch into the South Branch, especially during the spring tide from February 14 to 17, 2014. This is the famous SSO occurring during middle and spring tide in dry season.

In Exp 2, the seaward water flux on February 7 decreased and the landward water flux increased from February 4 to 6 and from February 8 to 14, and the salt flux from February 4 to 14 was landward, under strong northerly wind, which was distinctly greater than the result under climatic wind and residual water level conditions at open sea boundaries. Therefore, the strong northerly wind enhanced the SSO.

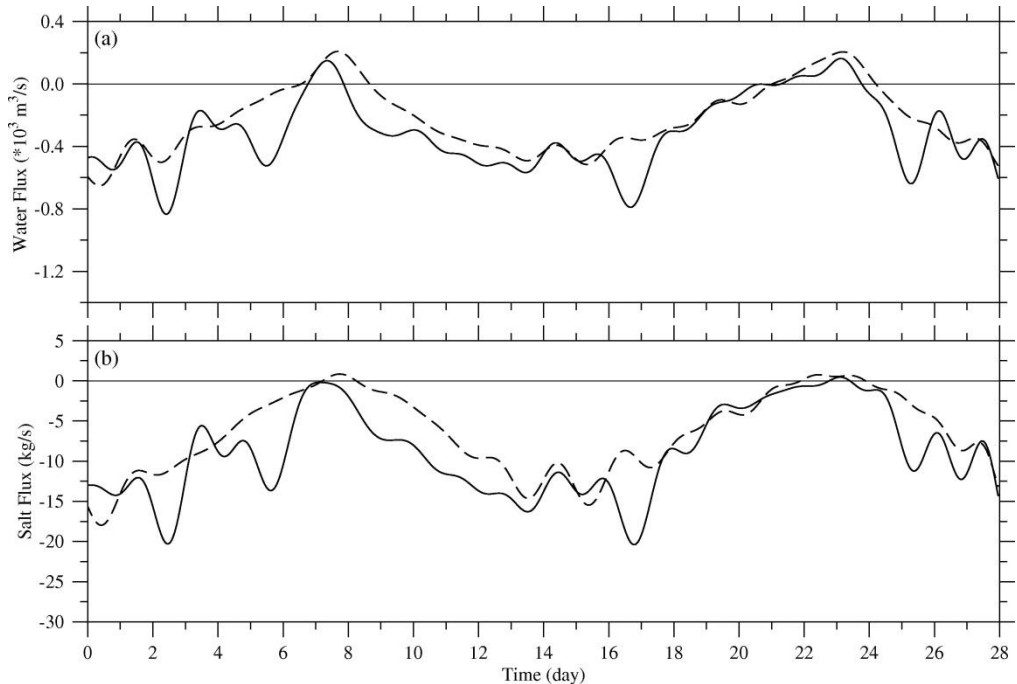

**Figure 11: Temporal variations in residual water flux (a) and salt flux (b) across Sec 2 in the North Branch in February 2014. Dashed line: Exp 1; solid line: Exp 2. A positive value represents seaward flux, and a negative value represents landward flux.**

A numerical experiment was designed in which the upper reaches of the North Branch were blocked (location labeled in Fig. 1) to further distinguish the contribution of SSO in the saltwater intrusion event. The distribution of time-averaged surface salinity from February 10 to 13, 2014 (Fig. 12a), and the temporal variations in salinity from February 8 to 22, 2014, at Chongxin station (Fig. 12b) show that the SSO was completely absent, while the salinity in the river mouth and near the Qingcaosha reservoir and at the Baozhen, Nanmen and Qingcaosh stations was almost identical to the results from Exp 2 (Fig. 7d, Fig. 3), indicating that the SSO had almost no contribution to the saltwater intrusion event in February 2014.

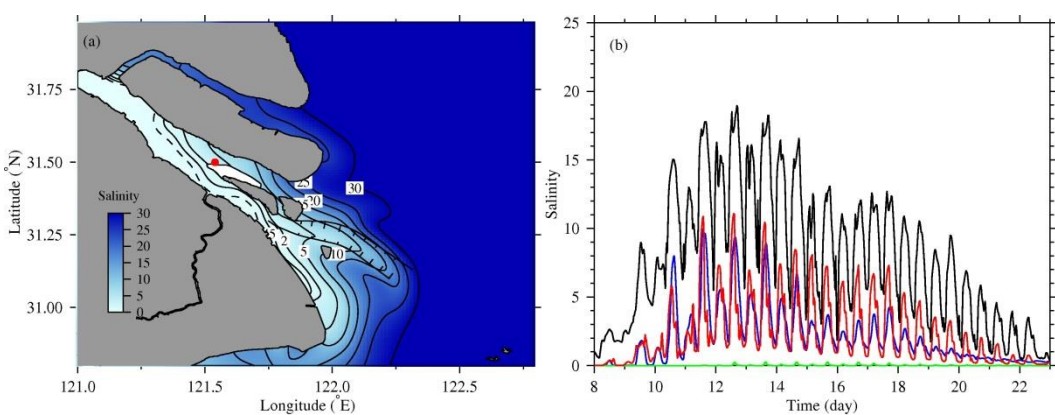

**Figure 12: Distribution of time-averaged surface salinity from February 10 to 13, 2014 (a); temporal variations in salinity from February 8 to 22, 2014, at hydrologic stations (b) if the upper reaches of the North Branch are blocked, and under realistic wind and residual water levels at the open sea boundaries. Black line: Baozhen; red line: Nanmen; green line: Chongxi; blue line: Qingcaosha.**

### 4.2 How does the wind affect the individual terms in the momentum equations?

The modeled individual terms in the momentum equations were output at site Mo in the North Channel (labeled in Fig. 1, water depth 9.62 m) to analyze how the wind affects these terms. The momentum along the river channel ($\xi$ direction) is:

$$\frac{\partial u}{\partial t} = -\frac{\partial u^2}{\partial \xi} - \frac{\partial uv}{\partial \eta} - \frac{\partial wu}{D \partial \sigma} + vf - g\frac{\partial \zeta}{\partial \xi} + \frac{g}{\rho_0}\frac{\partial D}{\partial \xi}\int_\sigma^0 \sigma\frac{\partial \rho}{\partial \sigma}d\sigma - \frac{gD}{\rho_0}\frac{\partial}{\partial \xi}\int_\sigma^0 \rho d\sigma + \frac{1}{D^2}\frac{\partial}{\partial \sigma}\left(K_m\frac{\partial u}{\partial \sigma}\right) + F_\xi \qquad (2)$$

The momentum across the river channel ($\eta$ direction) is:

$$\frac{\partial v}{\partial t} = -\frac{\partial uv}{\partial \xi} - \frac{\partial v^2}{\partial \eta} - \frac{\partial wv}{D \partial \sigma} - uf - g\frac{\partial \zeta}{\partial \eta} + \frac{g}{\rho_0}\frac{\partial D}{\partial \eta}\int_\sigma^0 \sigma\frac{\partial \rho}{\partial \sigma}d\sigma - \frac{gD}{\rho_0}\frac{\partial}{\partial \eta}\int_\sigma^0 \rho d\sigma + \frac{1}{D^2}\frac{\partial}{\partial \sigma}\left(K_m\frac{\partial v}{\partial \sigma}\right) + F_\eta \qquad (3)$$

where $\frac{\partial u}{\partial t}$ and $\frac{\partial v}{\partial t}$ represent the local variation, $-\frac{\partial u^2}{\partial \xi} - \frac{\partial uv}{\partial \eta} - \frac{\partial wu}{D \partial \sigma}$ and $-\frac{\partial uv}{\partial \xi} - \frac{\partial v^2}{\partial \eta} - \frac{\partial wv}{D \partial \sigma}$ are advection, $vf$ and $-uf$ indicate the Coriolis force, $-g\frac{\partial \zeta}{\partial \xi}$ and $-g\frac{\partial \zeta}{\partial \eta}$ represent the barotropic pressure gradient force, $\frac{g}{\rho_0}\frac{\partial D}{\partial \xi}\int_\sigma^0 \sigma\frac{\partial \rho}{\partial \sigma}d\sigma - \frac{gD}{\rho_0}\frac{\partial}{\partial \xi}\int_\sigma^0 \rho d\sigma$ and $\frac{g}{\rho_0}\frac{\partial D}{\partial \eta}\int_\sigma^0 \sigma\frac{\partial \rho}{\partial \sigma}d\sigma - \frac{gD}{\rho_0}\frac{\partial}{\partial \eta}\int_\sigma^0 \rho d\sigma$ are the baroclinic pressure gradient force, $\frac{1}{D^2}\frac{\partial}{\partial \sigma}\left(K_m\frac{\partial u}{\partial \sigma}\right)$ and $\frac{1}{D^2}\frac{\partial}{\partial \sigma}\left(K_m\frac{\partial v}{\partial \sigma}\right)$ are vertical turbulence viscosity, and $F_\xi$ and $F_\eta$ are horizontal turbulence viscosity in the $\xi$ and $\eta$ directions, respectively. Each output term in the momentum equations was vertically averaged, and the vertical turbulence viscosity $\frac{1}{D^2}\frac{\partial}{\partial \sigma}\left(K_m\frac{\partial u}{\partial \sigma}\right)$ and $\frac{1}{D^2}\frac{\partial}{\partial \sigma}\left(K_m\frac{\partial v}{\partial \sigma}\right)$ equals $\frac{\tau_{a\xi}-\tau_{b\xi}}{D}$, $\frac{\tau_{a\eta}-\tau_{b\eta}}{D}$, in which $\tau_{a\xi}$, $\tau_{a\eta}$ and $\tau_{b\xi}$, $\tau_{b\eta}$ are the wind stress and bottom friction force in the $\xi$ and $\eta$ direction, respectively. The tidal fluctuations in the vertically averaged terms were filtered using the moving average method.

In Exp 1, the residual water movement along the North Channel was mainly determined by the seaward barotropic pressure gradient force and Coriolis force (Fig. 13a). The residual barotropic pressure gradient force was positive (seaward), induced mainly by river discharge, varied with spring and neap tide, and had a smaller value from February 7 to 9, 2014 (during the neap tide, Fig. 6). The seaward runoff produced a negative Coriolis force. The baroclinic pressure gradient force was very small before February 10 because the saltwater intrusion was very weak at the model output site, then became larger with the increasing tide. This term is always landward in estuaries because the saline water is downstream. Across the channel, the

residual water movement was also mainly determined by the barotropic pressure gradient force and Coriolis force (Fig. 13b).

The barotropic pressure gradient force was positive (northward) because the water level is higher on south side of the channel than on north side due to the effect of the Coriolis force on runoff. The baroclinic pressure gradient force was negative because salinity was higher on the north side than on the south side due to the effect of the Coriolis force on the flood current that brought saline water into the estuary.

  The results of Exp 2 were considerably different from those in Exp 1 (Fig. 13c, d). The persistent and strong wind induced

greater southward wind stress (red solid line in Fig. 13d) and produced landward Ekman transport and a rising water level, which significantly reduced the seaward barotropic pressure gradient force, especially from February 7 to 8, 2014. At this time, the barotropic pressure gradient force was very small, even negative (landward), and thus, the net water transport was landward, greatly enhancing the saltwater intrusion in the North Channel while distinctly increasing the landward baroclinic pressure gradient force (black dashed line in Fig. 13c).

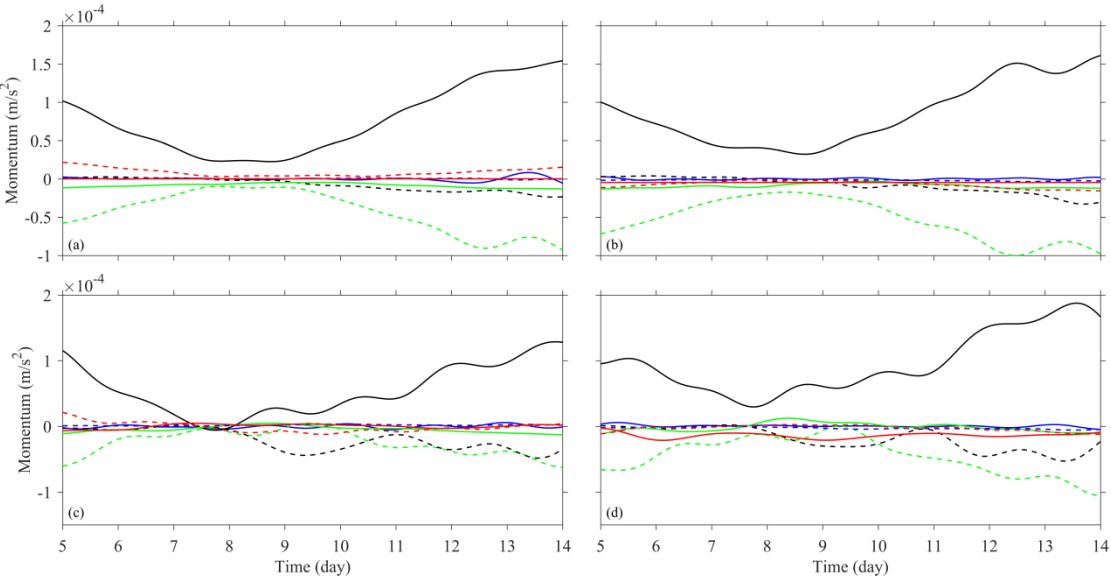


**Figure 13: Temporal variations in the vertically and time averaged terms in the momentum equations along the channel (left panel, a and c) and across the channel (right panel, b and d) from February 5 to 13, 2014, in Exp 1 (upper panel) and Exp 2 (lower panel) at site Mo in the North Channel. Red solid line: wind stress at river surface; red dashed line: bottom friction force; black solid line: barotropic pressure gradient force; black dashed line: baroclinic pressure gradient force; green solid line: Coriolis**

**force; green dashed line: advection; blue solid line: local variation; blue dashed line: horizontal turbulence viscosity.**

## 4.3 The relationship between residual water level and wind

  The residual (mean) water level in the Changjiang Estuary is closely correlated with the wind. The water level at Sheshan and Luchaogang stations rose during the northerly wind and dropped during the southerly wind in February 2014 (Fig. 4d).

The water level rise of more than 50 cm corresponded to the strong northerly wind. The water level was approximately 20 cm (Fig. 5b) under climatic wind in February and 30 cm (Fig. 9b) under persistent and strong northerly wind in February 2014. At Baozhen, Sheshan and Luchaogang stations in February 2014, the modeled water level under climatic wind was lower than that under strong northerly wind, especially from February 6 to 10, 2014, when the difference reached 10-30 cm (Fig. 6). Therefore, the stronger the northerly wind is, the higher the water level in the Changjiang Estuary and adjacent sea.

**4.4 The generation of Ekman transport and the resulting horizontal circulation**

    The northerly wind produces southward current along the China coast and pushes water landward, transported by the Coriolis force, resulting in a high water level along the coast. In the Changjiang Estuary, this dynamic mechanism causes a pure wind-driven horizontal circulation that flows into the North Channel and out of the South Channel (Fig. 14a). The wind-driven estuarine current can enhance saltwater intrusion in the North Channel and weaken it in the South Channel. Previous 340 studies revealed the dynamic mechanism of northerly wind on the saltwater intrusion by the pure wind-driven current in the estuary (Wu et al., 2010; Li et al, 2012). In this study, the horizontal estuarine circulation was a total (net) circulation forced by the river discharge, tide and persistent and strong northerly wind (Fig. 7c) and was not a purely wind-driven circulation, which surpassed the strong seaward runoff from February 9 to 13, 2014 (Fig. 8a). This result was unexpected and surprising and was found for the first time.

Numerical experiments were performed with different northerly wind speeds to ascertain the influence of the wind on the water flux at sec1 in the landward transport in the North Channel. Except for the wind, the other dynamic factors were the same as in Exp 2. The results show that the net water flux is seaward when the wind speed is less than 8 m/s, approximately zero when the wind speed is 10 m/s, and landward when the wind speed is greater than 10 m/s (Fig. 14b). The landward water flux enhances with an increasing northerly wind. Therefore, northerly wind with speed of more than 10 m/s can be considered 350 a strong wind that causes net water transport landward in the North Channel.

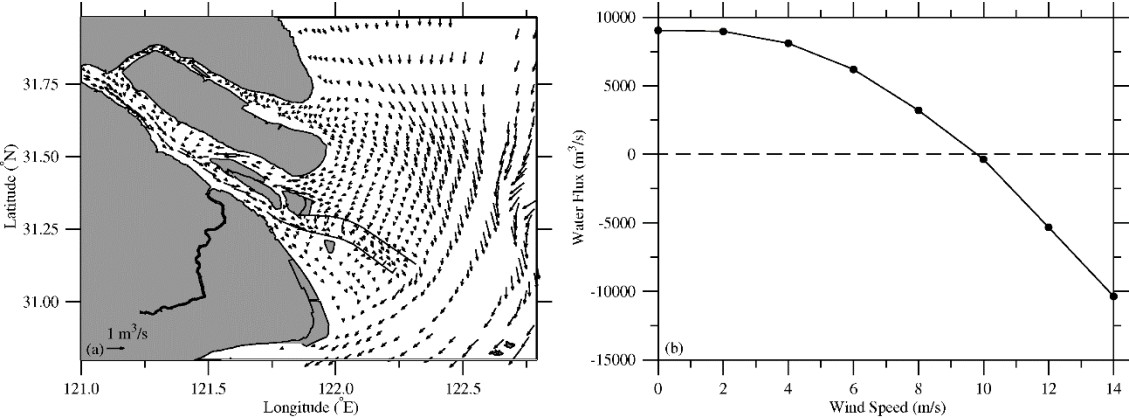

**Figure 14: Distribution of pure wind-driven current time averaged from February 10 to 13, 2014, modeled under the real wind (a). Relationship between water flux across sec1 in the North Channel and northerly wind speed (b). Except for the wind, the other dynamic factors in the numerical experiments are the same as in Exp 2.**

**4.5 How long does the northerly wind induce severe saltwater intrusions?**

In winter, frequent cold fronts pass over the Changjiang Estuary and bring strong northerly winds, enhancing saltwater intrusion (Li et al., 2012). For example, the relatively strong saltwater intrusions from February 15 to 18, 2011, and from February 23 to 26, 2017, were caused by the passing cold fronts. No extremely severe saltwater intrusion event occurred because the strong northerly wind induced by the cold fronts lasted only 1-2 days. How long does the northerly wind induce severe saltwater intrusion? Based on the persistent and strong northerly wind process in February 2014, numerical experiments were conducted, and the results indicate that if a strong northerly wind lasts 2 days, saltwater intrusion is weak (Fig. 15), which is similar to the case of an ordinary cold front passage; if a strong northerly wind lasts 4 days, saltwater intrusion is stronger than normal, similar to the case of a stronger cold front passage. It can been seen that 4 days of strong wind can induce the higher than normal salinity after 8 days because the strong wind Ekman transport brought the salinity front in the sandbar area upstream and closer to the Baozhen station and then moved upstream and downstream with the oscillation of flood and ebb currents for 8 days after the strong northerly wind became northerly wind with speed of 5 m/s. If a strong northerly wind lasts 6 days, saltwater intrusion becomes severe; if a strong northerly wind lasts 8 days, the salinity is dramatically increased, the maximum salinity approaches the real salinity, and the saltwater intrusion becomes extremely severe. Therefore, the longer the strong northerly wind lasts, the more severe the saltwater intrusion is.

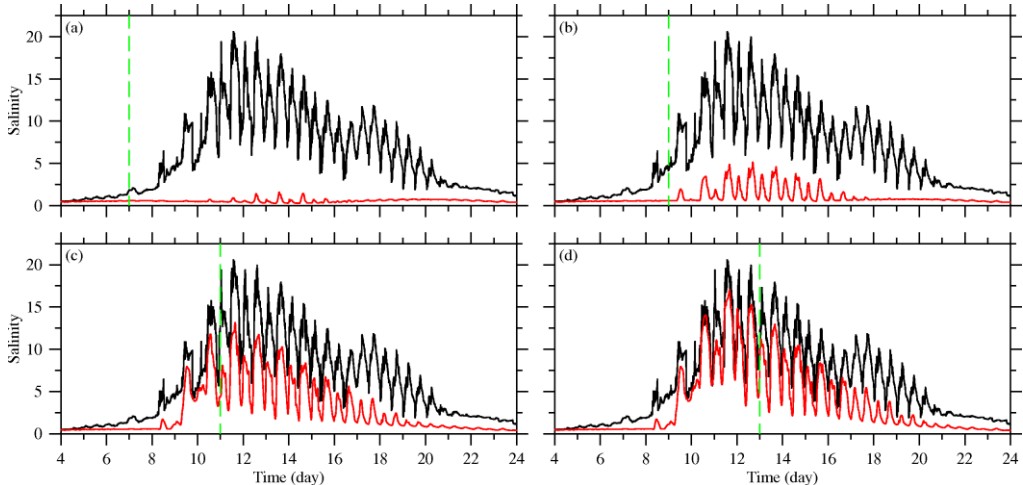

**Figure 15: Temporal variation in observed (black line) and modeled (red line) salinity at Baozhen station from February 4 to 24, 2014. The green dashed lines indicate the time before it the wind was realistic and after it was set to 5 m s⁻¹.**

**5 Conclusions**

An extremely severe saltwater intrusion event in February 2014 occurred in the Changjiang Estuary under normal river discharge conditions and caused a serious threat to water safety in Shanghai. No catastrophic saltwater intrusion of this magnitude has occurred since salinity in the estuary has been recorded. Our findings show that this extremely severe saltwater intrusion event was caused by a persistent and strong northerly wind, which produced a strong landward net transport and

abnormal high water level along the coast of China and drove substantial landward water transport to form a horizontal estuarine circulation that flowed into the North Channel and out of the South Channel. This landward net water transport overcame seaward runoff and brought very large amounts of highly saline water into the upper reaches of the North Channel, which seriously threatened the water intake of the Qingcaosha Reservoir.

To further reveal the dynamic mechanism of the event, numerical experiments were run, and the results were discussed. The strong northerly wind can enhance the SSO, but the SSO had almost no contribution to the saltwater intrusion event in February 2014. The individual terms in the momentum equations showed that the persistent and strong wind induced larger southward wind stress and produced landward Ekman transport and a rising water level, which significantly reduced the seaward barotropic pressure gradient force and pushed the net water transport landward, greatly enhancing the saltwater intrusion in the North Channel while distinctly increasing the landward baroclinic pressure gradient force. The stronger the northerly wind is, the higher the water level in the Changjiang Estuary and adjacent sea. The strong northerly wind can produce a pure wind-driven horizontal circulation that flows into the North Channel and out of the South Channel, which can enhance saltwater intrusion in the North Channel and weaken it in the South Channel. In this study, the horizontal estuarine circulation was a total (net) circulation induced by the persistent and strong northerly wind, which surpassed the strong seaward runoff. This effect represents a novel and unexpected finding. Only a northerly wind with speed of greater than 10 m/s can cause net landward water transport in the North Channel. A northerly wind lasting 8 days can produce extremely severe saltwater intrusion in the Changjiang Estuary. With more frequent and stronger northerly winds caused by climate change, greater attention should be paid to extremely severe saltwater intrusions and freshwater safety in the Changjiang Estuary, where the Qingcaosha Reservoir provides water from the estuary for 13 million people in Shanghai.

*Code Availability.* The codes can be made available from the corresponding author upon reasonable request.

*Data Availability.* The data underlying the findings of this article can be accessed at https://doi.org/10.6084/m9.figshare.c.5114444.v1.

*Author contributions.* Jianrong Zhu analyzed the dynamics of the extremely severe saltwater intrusion in the Changjiang Estuary. Xinyue Cheng simulated the persistent and strong northerly wind with the Weather Research Forecasting (WRF) model and analyzed the model results. Linjiang Li simulated the extremely severe saltwater intrusion in the Changjiang Estuary. Hui Wu simulated the residual water level with the large domain model encompassing the Bohai Sea, Yellow Sea and East China Sea. Jinghua Gu measured the salinity at the hydrological stations. Hanghang Lyu validated the saltwater intrusion model.

*Competing interests.* The authors declare that they have no conflict of interest.

*Acknowledgments.* The authors thank the Shanghai Hydrology Administration for providing the observed water level data in the Changjiang Estuary.

*Financial support.* This work was supported by the National Natural Science Foundation of China (41676083, 41476077) and Major Program of Shanghai Science and Technology Committee (14231200402).

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
