# Peer review of "Dynamic mechanism of an extremely severe saltwater intrusion in the Changjiang Estuary in February 2014"

_Hydrology and Earth System Sciences, 2020_

## Referee Comment (RC1) · Anonymous Referee #1 · 12 Jul 2020

In this manuscript, the authors investigated an extremely severe saltwater intrusion in February 2014 in the Changjiang Estuary by means of a three-dimensional numerical model. It was shown that the catastrophic saltwater intrusion leading to a cut off freshwater input for 23 days into the Qingcaosha Reservoir can be primarily attributed to a persistent and strong northerly wind, which drove substantial landward Ekman transport to form a horizontal estuarine circulation. The paper is generally well written. However, there are still several major concerns that should properly addressed in order to improve the quality of this manuscript. In particular, the underlying mechanism of extremely severe saltwater intrusion with regard to the persistent and strong northerly wind should be detailed in the discussion part.

1) Section 2.1: It can be seen from observed salinity (in Figure 2a) that at normal condition the salinity at the upstream Chongxi station is higher than other stations, while during the abnormal condition (i.e., the studied case) the maximum salinity concentration appears at Baozhen station. I would suggest the authors to clarify the difference between these two conditions since the underlying mechanism for saltwater intrusion may be completely different. For instance, to what extent the spilling of saltwater from the North Branch into the South Branch may impact the dynamics of saltwater concentration during this abnormal condition when compared with the normal condition.

2) Section 2.1: With regard to this section, I would suggest to move the analysis of the observed data into "Results" section 3.

3) Calibration and Validation of the numerical model: Although the saltwater intrusion model has been extensively calibrated and validated, I would suggest the authors to present the calibration for the salinity along the channel (i.e., Baozhen, Nanmen, Chongxi and Qingcaosha). For the time being, we can only see the reproduction of the salinity in Baozhen station, while the performances in other available stations are unknown. In addition, it would be better to clarify the basic calibration parameters used in the numerical model in the manuscript.

4) Section 4: In principle, the discussion part should systematically discuss the potential mechanism of saltwater intrusion with regard to the persistent and strong northerly wind. How the wind affects the individual terms in the momentum equation? The relationship between residual water level rise and wind? The generation of Ekman transport and the resulted horizontal circulation etc. In current form, the discussion part is more or less the same as the conclusions part since we see an identical sentence "With more frequent persistent and strong northerly wind caused by climate change, more attention should be paid to extremely severe saltwater intrusion events and freshwater safety in the Changjiang Estuary because the Qingcaosha Reservoir takes water from the Changjiang Estuary for the 13 million people in Shanghai." in both sections.

---

## Referee Comment (RC2) · Anonymous Referee #2 · 25 Jul 2020

The authors looked into the salinity intrusion issue in the Changjiang Estuary that occurred in February 2014 and demonstrate that the persistent and strong northerly wind is the cause of this issue. It is significant for studying the water safety of freshwater resources.

However, I have two concerns. 1, According to Figure 5b, the realistic wind can be a cause. However, what is the role of the spring tide (The period from 10 Feb to 18 Feb 2014 according to the lunar calendar) for the intrusion during this issue?

2, Had some issues also happened in the near Hangzhou Bay? Since the wind also causes a rise of residual water level there (Figures 4, 7, 8). If yes please add some details; if no what caused the different issues between the Changjiang Estuary (saltwater intrusion) and Hangzhou Bay (no issue) since the wind would be equal for these two.

Minor issue: 1, Line 239, Page 14, Check the name.

---

## Referee Comment (RC3) · Anonymous Referee #3 · 29 Jul 2020

**Review of "Dynamic mechanism of extremely severe saltwater intrusion in the Changjiang Estuary occurred in February 2014" by Zhu et al.**

In this manuscript, authors simulated the severe saltwater intrusion in the Changjiang Estuary in February 2014 based on numerical model. They believed that the severe saltwater intrusion was induced by persistent strong northerly winds. The manuscript is not well written, and has many issues and doubts which are as follows.

**Major issues:**

1. There are many studies about the influence of winds on saltwater intrusion in the estuary. But authors only mentioned two papers (Xue et al., 2009; Li et al., 2012) about the Changjiang Estuary. Even about the Changjiang Estuary, there are not only two papers.

2. About the strong wind event, it should be defined such as wind speed and duration. Authors said that from February 5 to 14, 2014 a persistent and strong northerly wind occurred lasting ten days. And they presented that only a strong northerly wind lasting 8 days can produce a severe saltwater intrusion in the Changjiang Estuary. But seen from plot c of Figure 2, on 5-6 the wind directions were southerly and easterly, and the winds seemed not strong. On 12-14, the winds were not strong as well. Authors should show what magnitude of wind event could induce severe saltwater intrusion. In plot c of Figure 2, the curve of wind speeds should be added. Thus the magnitude of winds can be seen clearly. What kind of data was used in plot c, instantaneous value, 2 minutes average, or maximum in a gust of wind? They should be presented clearly. In addition, the weather station locates inside the estuary near the mouth. The wind direction at this station may be different from the sea.

3. This severe saltwater intrusion event is strange. The peak salinity at Baozhen and Nanmen stations reached 20.1 and 12.4 respectively. But why did the salinity only reach 8.6 at Qingcaosha? The location of Qingcaosha is close to Baozhen, and downstream of Nanmen. It can be seen from plot a of Figure 2 that salinity at Qingcaosha was much lower

than Baozhen. In addition, salinity at Chongxi station is from the North Branch. Before 7 February salinity was high at Chongxi, very low even close to zero at other stations. But during the severe event salinity was very high at other stations, but low at Chongxi station. These need explanation.

4. Authors presented that the water level rose distinctly at the coast during the event (Figure 2, line 91). Why did the water level inside the estuary not rise obviously (Figure 5, lines 133-135)? Authors said that the water level inside the river mouth was mainly determined by tide and river discharge. This needs explanation. Seen from plot b of Figure 4 and plot b of Figure 7, the water level rise inside the estuary was much larger than the coast. This is inconsistent with authors' expression and observations at Baozhen. The water level rises in figure 7 and figure 8 seem the same both for 10-13 February. In addition, about plot d in Figure 2, how was the water level rise obtained or how did authors calculate the water level rise? This should be presented clearly in methods section.

5. The main work of this manuscript is modeling of salinity and water level during the severe event. But authors only presented the results at Baozhen station (Figure 5). The results at other stations should be shown as well. Figures 6-8 only present the time-averaged results on 10-13 February.

6. The dynamic mechanism of the severe saltwater intrusion event is the objective of this manuscript. The manuscript proposed the mechanism: landward Ekman transport forms a horizontal estuarine circulation that flowed into the North Channel and out of the South Channel. This mechanism or result is not new. It has been presented in authors' previous work (Wu Hui, Zhu Jianrong, Choi Byung Ho, 2010. Links between saltwater intrusion and subtidal circulation in the Changjiang Estuary: a model-guided study. Cont. Shelf Res. 30 (17): 1891–1905.). But authors did not mention this work. This reference did not occur in the manuscript as well. The strong winds were not persistent for very long time. And the wind directions were not always northerly, even southerly in some periods. Why did the severe saltwater intrusion last

23 days? This is the question the manuscript should answer. About the results presented in discussion part and figure 9, it is doubtful. About plot b, can two-day strong winds induce the higher than normal salinity in after 8 days? About plots c and d, there are similar doubts. What is the mechanism of this? This needs detailed explanation. In addition, many studies mentioned that water withdrawal between Datong and estuary could increase saltwater intrusion in dry season. Is there possibility that during the severe event water withdrawal downstream Datong was large contributing to this event as well?

7. The structure of the manuscript is strange. In section 2.1 observed data, authors introduced the severe saltwater intrusion events, which should be moved to introduction section or results section. In section 3 results, section 3.1 is not necessary. The results under normal situations and special situations can be compared in order to show the difference. But the result under normal situation is not the important results for the objective of the manuscript. In other words, it is not necessary presented separately. In addition, the discussion part is too simple.

8. Some presentations or data in the manuscript are unreliable. Besides some mentioned above examples are as follows.

(1) About the data source, in section 2.1 (observed data), authors said that the observed data was conducted by State Key Laboratory of Estuarine and Coastal Research, East China Normal University. But in acknowledgements part authors said that the observed data was provided by Shanghai Hydrology Administration.

(2) Page 3, line 57, "... caused by very low river discharge of approximately 7000 and 8000 $m^3$ $s^{-1}$ lasting three mouths (mouths should be months), respectively". In this sentence, "8000 $m^3$ $s^{-1}$ in 1999" and "lasting three months" are not correct. In dry season of 1979, river discharges in January and February were really very low between 7000 and 8000 $m^3$/s at Datong station. In March the monthly mean discharge was more than 10000 $m^3$/s during which severe saltwater intrusion also occurred. In 1999 the monthly mean river discharges at Datong were all larger than 9000 $m^3$/s. In February extremely severe saltwater intrusion occurred as well inducing continuous 25 days of

unsuitable drinking water at Chenhang Reservoir upstream of Qingcaosha, during which discharge was 9110 m$^3$/s.

**Minor issues:**

1. In page 2, lines 30-31, what is the maximum spring tide and minimum neap tide? The proper expressions should be the maximum tidal range of spring tide and minimum tidal range of neap tide. Or, the maximum tidal range and minimum tidal range are enough.

2. In page 3, lines 49-50, "which was the largest estuarine reservoir in the world, the Qingcaosha Reservoir, was built.". There is syntax error in this sentence.

3. In the last paragraph of introduction, authors said that an extremely severe saltwater intrusion event in February 2014 occurred, and this is a catastrophic event never occurred. But we did not see how severe and catastrophic. The compare between this event and historical severe events should be presented as well in order to show the severe magnitude.

4. About the river discharge used during modelling, did you consider the time required for water traveling from Datong to the estuary? Usually the discharges several days in advance are used because Datong station is located more than 600 km upstream of the estuary.

5. Caption of Figure 2: Temporal variations in the measured data in February. Plot d does not present the measured data. The water level rises are calculated results.

6. Caption of Figure 3 is not clear. And there are syntax errors.

7. What is the climatic wind? This expression is strange.

8. Captions of Figures 5, 6, 8 are not clear.

---

## Author Comment (AC1) · 11 Aug 2020

Reply to anonymous referee #2

Dear reviewer,

We appreciate your valuable questions and comments on our manuscript. The manuscript has been carefully revised according to your advices, and the revisions are marked with blue (with red and green for the other two reviewers). The point-by-point answers to the questions and comments are listed as follows.

Your sincerely,

Jianrong Zhu, on behalves of the co-authors

**Comment 1:** According to Figure 5b, the realistic wind can be a cause. However, what is the role of the spring tide (The period from 10 Feb to 18 Feb 2014 according to the lunar calendar) for the intrusion during this issue?

**Reply:** As we can see from Figure 5b, the strongest saltwater intrusion occurred from February 11 to 14, 2014 in the middle tide, corresponding to lunar calendar 12 to 15. The tidal pattern of the middle tide was indicted by the measured tidal level at Baozhen hydrologic station in Figure 5a. The highest tidal level is in lunar calendar 18, not in 15 in the Changjiang Estuary. There are four tidal patterns in one neap-spring tide, neap tide, middle tide after neap tide, spring tide, and middle tide after spring tide. Each tidal pattern has approximately 4 days. Our study indicated that the saltwater intrusion event was caused by the persistent and strong northerly wind, which played a more important role on the saltwater intrusion event during neap tide

than spring tide.

**Comment 2:** Had some issues also happened in the near Hangzhou Bay? Since the wind also causes a rise of residual water level there (Figures 4, 7, 8). If yes please add some details; if no what caused the different issues between the Changjiang Estuary (saltwater intrusion) and Hangzhou Bay (no issue) since the wind would be equal for these two.

**Reply:** Thanks for your good comment. Hangzhou Bay is adjacent to the Changjiang Estuary, and is occupied by sea water all year round. The salinity is greatly higher than 0.45 (the standard salinity for drinking water) and there is no freshwater in the Hangzhou Bay. Therefore, there is no such wording of saltwater intrusion there. It's true that the strong northerly wind also caused a rise of residual water level, and caused rise of salinity in the Hangzhou Bay.

The Qiantang River connects the Hangzhou Bay, and saltwater intrusion often occurs in dry season. There is the most downstream water plant, Nanxing water plant, in the Qiantang River for water supply in Hangzhou (Figure R2-1). Because the water intake of the water plant is far away from the river mouth and the river depth is only 1-2 m, there is no same issue (severe saltwater intrusion in the Changjiang Estuary in February 2014) occurred there. There was no some issue in the North Branch of the Changjiang Estuary because its water depth is very shallow there. The reason is same for the Qiantang River. In the North Branch of the Changjiang Estuary, the landward wind-driven Ekman water transport was weaker, flowing along the north side and

flowing out along the south side only near the river mouth (Fig. 7c in the revised manuscript). However, the North Channel is deeper and wider and located on the north side of the South Branch, which is in favor of producing strong landward Ekman water transport in the North Channel, resulting in severe saltwater intrusion.

[Figure]

Figure R2-1 Satellite image of the Changjiang Estuary, Changjiang Bay and Qiantang River

**Minor issue**

1, Line 239, Page 14, Check the name.

**Reply: Thanks.** The name "Lijinag" has been changed to "Linjiang".

---

## Author Comment (AC2) · 11 Aug 2020

Reply to anonymous referee #3

Dear referee,

We appreciate your valuable questions and comments on our manuscript. The point-by-point answers to the questions and comments are listed as follows.

Your sincerely,

Jianrong Zhu, on behalves of the co-authors

**Major issues:**

**Major issue 1:** There are many studies about the influence of winds on saltwater intrusion in the estuary. But authors only mentioned two papers (Xue et al., 2009; Li et al., 2012) about the Changjiang Estuary. Even about the Changjiang Estuary, there are not only two papers.

**Reply:** Thanks for your good suggestion. Now we added the following papers the influence of winds on saltwater intrusion in the estuary.

Aristizabal, M. F., and Chant, R. J. (2015): An observational study of salt fluxes in Delaware Bay, Journal of Geophysical Research-Oceans, 120, 2751-2768, 2015.

Chen, S. N., and Sanford, L. P.: Axial wind effects on stratification and longitudinal salt transport in an idealized, partially mixed estuary, Journal of Physical Oceanography, 39(8), 1905-1920, 2009.

Duran-Matute, M., Gerkema, T., and Sassi, M. G.: Quantifying the residual volume transport through a multiple‐inlet system in response to wind forcing: The

case of the western Dutch Wadden Sea, Journal of Geophysical Research: Oceans, 12, 8888-8903, 2016.

About the Changjiang Estuary, the following papers the influence of winds on saltwater intrusion were added.

Wu H., Zhu J. R., and Choi B. H.: Links between saltwater intrusion and subtidal circulation in the Changjiang Estuary: a model-guided study, Cont. Shelf Res., 30, 1891-1905, 2010.

Zhang E, Gao S, Savenije H. H. G, Sic C, and Cao S: Saline water intrusion in relation to strong winds during winter cold outbreaks: North Branch of the Yangtze Estuary, Journal of Hydrology, 574: 1099-1109, 2019.

Ding L., Dou X. P., Gao X. Y., Jiao J., and Hu J.: Response of salinity intrusion to winds in the Yangtze Estuary, Proceedings of 5th International Conference on Hydraulic Engineering, CHE2017, 241-247, 2017.

**Major issue 2:** About the strong wind event, it should be defined such as wind speed and duration. Authors said that from February 5 to 14, 2014 a persistent and strong northerly wind occurred lasting ten days. And they presented that only a strong northerly wind lasting 8 days can produce a severe saltwater intrusion in the Changjiang Estuary. But seen from plot c of Figure 2, on 5-6 the wind directions were southerly and easterly, and the winds seemed not strong. On 12-14, the winds were not strong as well. Authors should show what magnitude of wind event could induce severe saltwater intrusion. In plot c of Figure 2, the curve of wind speeds should be

added. Thus the magnitude of winds can be seen clearly. What kind of data was used in plot c, instantaneous value, 2 minutes average, or maximum in a gust of wind? They should be presented clearly. In addition, the weather station locates inside the estuary near the mouth. The wind direction at this station may be different from the sea.

**Reply:** Thanks for your good question. There is really no definition of strong wind causing severe saltwater intrusion in the Changjiang Estuary. From the observed and simulated wind, and results of numerical experiments for wind effect on saltwater intrusion in the estuary, wind with speed of greater than 10 m/s and lasting 8 days can be called strong wind.

We checked the wind in plot c of Figure 2, the wind was southerly on 3 and easterly on 4, and was northeasterly on 5 with speed of approximately 10 m/s. The mean wind speed on 12-13 was approximately 10 m/s, and was only 5 m/s on 14. In fact, the measured wind at the weather station on the Chongming eastern shoal was just used to describe the phenomenon of persistent and strong wind, and was not used to simulate the severe saltwater intrusion event. The actually used wind in the model was the one simulated by the WRF, which wind speed adjacent sea near the Changjiang River mouth was stronger than on the Chongming eastern shoal (Figure R3-1). Now the sentence "a persistent and strong northerly wind from February 5 to 14, 2014, lasting ten days" was changed to "a persistent and strong northerly wind from February 5 to 13, 2014, lasting nine days".

In plot c of Figure 2, the curve of wind speeds has been added (Figure R3-2).

This figure was enlarged to be seen clearly the wind vector and wind.

The wind data used in plot c was 2 minutes average one, and was illustrated in the revised manuscript.

Yes, the weather station locates inside the estuary near the mouth, and the wind direction at this station was somewhat different from the sea (seen in Figure R3-1 and Figure R3-2). It needs to be emphasized again that the actually used wind in the model was the one simulated by the WRF, which wind speed on the sea was stronger than the one on the Chongming eastern shoal.

[Figure]

Figure R3-1 Temporal variations in wind vector (a) and wind speed (b) simulated by the WRF model off the Subei coast in February 2014.

[Figure]

Figure R3-2 Temporal variations in the measured river discharge at Datong station (a), wind vector (b) and wind speed (c) at WS, and water level rise obtained by subtracting the data in the tide table from the measured water level at Sheshan station (black line) and Luchaogang station (red line) (d) in February 2014 in February 2014.

**Major issue 3:** This severe saltwater intrusion event is strange. The peak salinity at Baozhen and Nanmen stations reached 20.1 and 12.4 respectively. But why did the salinity only reach 8.6 at Qingcaosha? The location of Qingcaosha is close to Baozhen, and downstream of Nanmen. It can be seen from plot a of Figure 2 that salinity at

Qingcaosha was much lower than Baozhen. In addition, salinity at Chongxi station is from the North Branch. Before 7 February salinity was high at Chongxi, very low even close to zero at other stations. But during the severe event salinity was very high at other stations, but low at Chongxi station. These need explanation.

**Reply:** It's a good question, and is exactly what this manuscript wants to reveal. From the longitudinal view, the extremely saltwater intrusion in February 2014 was came from the downstream sea, consequently the peak salinity at Baozhen, Nanmen and Chongxi stations was 20.1, 12.4 and 4.0 during the event, respectively.

From the transverse perspective, although the location of Qingcaosha is close to Baozhen and downstream of Nanmen, the salinity only reached 8.6 at Qingcaosha, and was much lower than Baozhen. This was because the saltwater water was brought by the flood current and turned to right under the effect of Coriolis force that made the salinity was higher on the north side than on the south side of the North Channel. The simulated salinity distribution showed the same phenomenon. No matter in normal condition or in the abnormal condition (i.e., the studied case), this transverse difference of salinity was same.

At normal condition with climatic wind in dry season, there exits the saltwater-spill-over (SSO) from the North Branch into the South Branch in spring tide that results in the salinity at Chongxi station is higher than at Nanmen and Baozhen stations. This is the most feature of saltwater intrusion in the Changjiang estuary, and is a well-known phenomenon, that makes salinity is high at Chongxi station than at Nanmen and Baozhen stations. While during the abnormal condition with persistent

and strong northerly wind in February 2014, the opposite was occurred. This is what we want to reveal.

**Major issue 4:** Authors presented that the water level rose distinctly at the coast during the event (Figure 2, line 91). Why did the water level inside the estuary not rise obviously (Figure 5, lines 133-135)? Authors said that the water level inside the river mouth was mainly determined by tide and river discharge. This needs explanation. Seen from plot b of Figure 4 and plot b of Figure 7, the water level rise inside the estuary was much larger than the coast. This is inconsistent with authors' expression and observations at Baozhen. The water level rises in figure 7 and figure 8 seem the same both for 10-13 February. In addition, about plot d in Figure 2, how was the water level rise obtained or how did authors calculate the water level rise? This should be presented clearly in methods section.

**Reply:** Thanks. The two different concepts of water level and water level rise need to be distinguished here. The water level rise at Sheshan and Luchaogang stations in Figure 2d was almost same as the one in Figure 5a. We said that the water level at Baozhen station inside the river mouth was mainly determined by tide and river discharge because the temporal variations in the observed and modeled water level was roughly same.

In plot b of Figure 4 and plot b of Figure 7, the time-averaged water level from February 10 to 13, 2014 was shown, not the water level rise. The mean water level was much higher in inside the estuary than the coast due to the river discharge.

The residual water level from February 10 to 13, 2014 in figure 7b and figure 8a was the same one, wad duplicated. These two figures were redrew, plot a of Figure 8 was deleted, and the residual surface current was moved to plot b of Figure 7 ( Figure R3-3).

[Figure]

Figure R3-3 Distributions of the temporally averaged wind field from February 7 to 14, 2014, as simulated by the WRF model (a), and the time-averaged water level and surface current from February 10 to 13, 2014, as simulated by the model encompassing the Bohai Sea, Yellow Sea and East China Sea (b).

About plot d in Figure 2, the water level was provided by Shanghai Hydrology Administration, and the water level rise was obtained by subtracting the data in the tide table from the measured water level value. Those were illustrated in the revised manuscript.

**Major issue 5:** The main work of this manuscript is modeling of salinity and water level during the severe event. But authors only presented the results at Baozhen station (Figure 5). The results at other stations should be shown as well. Figures 6-8 only present the time-averaged results on10-13 February.

**Reply:** It's a good suggestion. Now we presented the measured results of salinity at Baozhen, Nanmen, Chongxi and Qingcaosha staions, and of water level at Baozhen, Sheshan and Luchaogang staions (Figure R3-4, and Figure R3-5). The model was validated for salinity and water level (reviewer 1 suggested).

[Figure]

Figure R3-4 Temporal variations in salinity in February 2014 at hydrologic stations. a:

Baozhen station; b: Nanmen station; c: Chongxi station; c: Qingcaosha station. Black

line: measured salinity; blue line: simulated salinity under climatic wind and residual

water level conditions at open sea boundaries; red line: simulated salinity under a

realistic wind and residual water levels at the open sea boundaries. The dashed green

line represents salinity of 0.45, which is the standard for drinking water.

[Figure]

Figure R3-5 Temporal variations in water level in February 2014 at hydrologic

stations. a: Baozhen station; b: Sheshan station; c: Luchaogang station. Black line:

measured data; blue line: simulated data under climatic wind and residual water level

conditions at open sea boundaries; red line: simulated data under a realistic wind and

residual water levels at the open sea boundaries.

**Major issue 6:** The dynamic mechanism of the severe saltwater intrusion event

is the objective of this manuscript. The manuscript proposed the mechanism:

landward Ekman transport forms a horizontal estuarine circulation that flowed into the North Channel and out of the South Channel. This mechanism or result is not new. It has been presented in authors' previous work (Wu Hui, Zhu Jianrong, Choi Byung Ho, 2010. Links between saltwater intrusion and subtidal circulation in the Changjiang Estuary: a model-guided study. Cont. Shelf Res. 30 (17): 1891–1905.). But authors did not mention this work. This reference did not occur in the manuscript as well. The strong winds were not persistent for very long time. And the wind directions were not always northerly, even southerly in some periods. Why did the severe saltwater intrusion last 23 days? This is the question the manuscript should answer. About the results presented in discussion part and figure 9, it is doubtful. About plot b, can two-day strong winds induce the higher than normal salinity in after 8 days? About plots c and d, there are similar doubts. What is the mechanism of this? This needs detailed explanation. In addition, many studies mentioned that water withdrawal between Datong and estuary could increase saltwater intrusion in dry season. Is there possibility that during the severe event water withdrawal downstream Datong was large contributing to this event as well?

**Reply:** This is a really misunderstanding. Wu et al. (2010) did presented a horizontal estuarine circulation that flowed into the North Channel and out of the South Channel, but is was pure wind-driven current under the northerly wind of 7 m/s (Figure 10 in their paper). Li et al. (2012) also presented a pure wind-driven current (Figure 14 in their paper). This pure wind-driven current was used to explain how the northerly wind effected the saltwater intrusion in the Changjiang Estuary. The pure

wind-driven estuarine current can enhance saltwater intrusion in the North Channel, and weaken it in the South Channel. In this study, the horizontal estuarine circulation was a total (net) circulation induced by the river discharge, tide and persistent and strong northerly wind, was not a pure wind-driven circulation, which surpassed the strong seaward runoff. This result was unexpected and surprising, and was found for the first time. The relevant illustration was added in the revised manuscript. The paper of Wu et al. (2010) was cited in the revised manuscript.

This rarely happened that the northerly wind with speed of great than 10 m/s and lasting more than 8 days in the Changjiang Estuary, but it did occurred (Figure R3-2), which caused the severe saltwater intrusion in February 2014, influencing the water intake of Qingcaosha reservoir for 23 days. If the wind directions were not always northerly, even southerly in some periods, the saltwater intrusion would be more severe and more serious impact on the Qingcaosha reservoir. This manuscript exactly answers why the extremely saltwater intrusion happened and what was the dynamic mechanism.

In Figure 9, the strong northerly wind lasts two, four, six and eight days in plot a, b, c and d, respectively. Two-day strong winds is in plot a, not in plot b. This is true that several days strong wind can induce the higher than normal salinity in after 8 days, because the strong wind Ekman transport brought the salinity front in sandbar area upstream and closer to the Baozhen station (model output site in Figure 9), and then move upstream and downstream with the oscillation of flood and ebb current for 8 days after the strong northerly wind became northerly wind with speed of 5 m/s. It

will take about 8 days to eliminate the impact of the upstream forward salinity front on the salinity at Baozhen station. This phenomenon was also shown in Figure 2a, 2c. The relevant illustration was added in the revised manuscript.

Yes, many studies pointed out that water withdrawal between Datong and the estuary could increase saltwater intrusion in dry season, but there was no possibility that the water withdrawal downstream Datong had large contributing to the severe saltwater intrusion event, because water withdrawal amount by the South-to-North Water Diversion Project, and water diversion and drainage along the river was approximately 1000 m$^3$/s, and the monthly mean river discharge was 11,500 m$^3$/s.

**Major issue 7:** The structure of the manuscript is strange. In section 2.1 observed data, authors introduced the severe saltwater intrusion events, which should be moved to introduction section or results section. In section 3 results, section 3.1 is not necessary. The results under normal situations and special situations can be compared in order to show the difference. But the result under normal situation is not the important results for the objective of the manuscript. In other words, it is not necessary presented separately. In addition, the discussion part is too simple.

**Reply:** Thanks for your good suggestion. Now we moved the section 2.1 observed data to the section 3 results in the revised manuscript, and deleted the title of 3.1 Climatic wind and residual water level conditions at open sea boundaries, and 3.2 Under a realistic wind in February 2014 and residual water levels at the open sea boundaries in the original manuscript.

According to the reviewer 1 and your suggestions, the discussion part has been greatly expanded. How much contribution the SSO had in the saltwater intrusion event, how the wind affects the individual terms in the momentum equations, the relationship between residual water level rise and wind, and the generation of Ekman transport and the resulted horizontal circulation were added and discussed in the revised manuscript.

**Major issue 8:** Some presentations or data in the manuscript are unreliable. Besides some mentioned above examples are as follows.

(1) About the data source, in section 2.1 (observed data), authors said that the observed data was conducted by State Key Laboratory of Estuarine and Coastal Research, East China Normal University. But in acknowledgements part authors said that the observed data was provided by Shanghai Hydrology Administration.

(2) Page 3, line 57, "... caused by very low river discharge of approximately 7000 and 8000 $m^3s^{-1}$ lasting three mouths (mouths should be months), respectively". In this sentence, "8000 $m^3s^{-1}$ in 1999" and "lasting three months" are not correct. In dry season of 1979, river discharges in January and February were really very low between 7000 and 8000 $m^3s^{-1}$ at Datong station. In March the monthly mean discharge was more than 10000 $m^3s^{-1}$ during which severe saltwater intrusion also occurred. In 1999 the monthly mean river discharges at Datong were all larger than 9000 $m^3s^{-1}$. In February extremely severe saltwater intrusion occurred as well inducing continuous 25 days of unsuitable drinking water at Chenhang Reservoir

upstream of Qingcaosha, during which discharge was 9110 $m^3s^{-1}$.

**Reply:** Thanks for pointing out the some incorrect expressions in the original manuscript.

(1) In section 2.1 (observed data), we modified the expression as: the observed salinity and wind data was conducted by State Key Laboratory of Estuarine and Coastal Research, East China Normal University, and the observed water level data was conducted by Shanghai Hydrology Administration. In acknowledgements part, we modified the expression as: the authors thank Shanghai Hydrology Administration providing the observed water level in the Changjiang Estuary.

(2) We checked and drew the variation processes of river discharge measured at Datong station in dry seasons of 1979 and 1999 (Figure R3-6). In dry season of 1979 from January 1 to March 15, the river discharge at Datong station was between 6000 and 9280 $m^3s^{-1}$ with mean value of 7485 $m^3s^{-1}$, lasting 2.5 months. In dry season of 1999 from January 1 to March 16, the river discharge at Datong station was between 7650 and 1,0900 $m^3s^{-1}$ with mean value of 9246 $m^3s^{-1}$, lasting 2.5 months.

Therefore, we modified the sentence "Historically, there have been two severe saltwater intrusion events in the Changjiang Estuary in the dry seasons of 1979 and 1999, which were caused by very low river discharge of approximately 7000 and 8000 $m^3s^{-1}$ lasting three months" to "Historically, there have been two severe saltwater intrusion events in the Changjiang Estuary in the dry seasons of 1979 and 1999, which were caused by very low river discharge of mean value of 7485 $m^3s^{-1}$ from January 1 to March 15, 1979, and 9246 $m^3s^{-1}$ from January 1 to March 16, 1999,

both lasting 2.5 months".

The word "mouths" has been changed to "months".

[Figure]

Figure R3-6 Temporal variations in river discharge measured at Datong station from December 1, 1978 to April 30, 1979 (a), and from December 1, 1998 to April 30, 1999 (b).

**Minor issues:**

**Minor issue 1:** In page 2, lines 30-31, what is the maximum spring tide and minimum neap tide? The proper expressions should be the maximum tidal range of spring tide and minimum tidal range of neap tide. Or, the maximum tidal range and minimum tidal range are enough.

**Reply:** Thanks. "the maximum spring tide" was changed to "the maximum tidal range", and "the minimum neap tide" was changed to "minimum tidal range".

**Minor issue 2:** In page 3, lines 49-50, "which was the largest estuarine reservoir in the world, the Qingcaosha Reservoir, was built". There is syntax error in this sentence.

**Reply:** "which was the largest estuarine reservoir in the world, the Qingcaosha Reservoir, was built" was changed to "that the largest estuarine reservoir in the world, the Qingcaosha Reservoir, was built".

**Minor issue 3:** In the last paragraph of introduction, authors said that an extremely severe saltwater intrusion event in February 2014 occurred, and this is a catastrophic event never occurred. But we did not see how severe and catastrophic. The compare between this event and historical severe events should be presented as well in order to show the severe magnitude.

**Reply:** The extremely severe saltwater intrusion event in February 2014 resulted in the continuous period of unsuitable drinking water reaching 23 days, and caused a serious threat to the water intake of the Qingcaosha Reservoir and water safety in Shanghai. So, we called the saltwater intrusion a catastrophic event. Historically, there were severe saltwater intrusions, such as in dry seasons of 1979 and 1999, but there was no reservoir in the Changjiang Estuary at that time. So, we only called these saltwater intrusions severe events. A catastrophic event refers to water security.

**Minor issue 4:** About the river discharge used during modelling, did you

consider the time required for water traveling from Datong to the estuary? Usually the discharges several days in advance are used because Datong station is located more than 600 km upstream of the estuary.

**Reply:** It's a good question. The west open boundary of the Changjiang River in the model is up to Datong station in the Changjiang River. So, it is not necessary to consider the time required for water traveling from Datong to the estuary.

**Minor issue 5:** Caption of Figure 2: Temporal variations in the measured data in February. Plot d does not present the measured data. The water level rises are calculated results.

**Reply:** Thanks. Now the caption of Figure 2 was changed to "Temporal variations in the measured river discharge at Datong station (a), wind vector (b) and wind speed (c) at WS, and water level rise obtained by subtracting the data in the tide table from the measured water level at Sheshan station (black line) and Luchaogang station (red line) (d) in February 2014.

**Minor issue 6:** Caption of Figure 3 is not clear. And there are syntax errors.

**Reply:** The caption of Figure 3 was rewritten.

Figure 2 Model grids of the Changjiang Estuary (a), and model grids of the Bohai Sea, Yellow Sea and East China Sea (b). Domains of the models (c); within the red line: the Changjiang Estuary model domain; within the green line: model domain of the Bohai Sea, Yellow Sea and East China Sea; the black dashed lines: the two-fold

nested WRF model domain.

**Minor issue 7:** What is the climatic wind? This expression is strange.

**Reply:** Climate wind means monthly mean wind for many years.

**Minor issue 8:** Captions of Figures 5, 6, 8 are not clear.

**Reply:** The captions of Figures 5, 6, 8 were rewritten. The Figures were reorganized and redrew, and the modified captions were presented in the revised manuscript.

---

## Author Comment (AC3) · 11 Aug 2020

Reply to anonymous referee #1

Dear referee,

We appreciate your valuable questions and comments on our manuscript. The point-by-point answers to the questions and comments are listed as follows.

Your sincerely,

Jianrong Zhu, on behalves of the co-authors

**Comment 1:** Section 2.1: It can be seen from observed salinity (in Figure 2a) that at normal condition the salinity at the upstream Chongxi station is higher than other stations, while during the abnormal condition (i.e., the studied case) the maximum salinity concentration appears at Baozhen station. I would suggest the authors to clarify the difference between these two conditions since the underlying mechanism for saltwater intrusion may be completely different. For instance, to what extent the spilling of saltwater from the North Branch into the South Branch may impact the dynamics of saltwater concentration during this abnormal condition when compared with the normal condition.

**Reply:** Thanks for your good comment. At normal condition with climatic wind in dry season, there exits the saltwater-spill-over (SSO) from the North Branch into the South Branch in spring tide that results in the salinity at Chongxi station is higher than at Nanmen and Baozhen stations. While during the abnormal condition with persistent and strong northerly wind in February 2014, the maximum salinity was greatly higher at Baozhen station than at Chongxi station. In order to quantitative analysis of impact of the SSO on the saltwater intrusion in the South Branch, the temporal variations in residual water flux and salt flux across sec2 (labeled in Figure 1) at the upper reaches of the North Branch in February, 2014 were added under climatic wind and real wind in February 2014. In addition, a numerical experiment in which the upper reaches of the North Branch is blocked was designed to distinguish how much effect of the SSO on saltwater intrusion in the South Branch under the real wind in February 2014. The above contents are added in the discussion section 4.1 in the

revised manuscript.

In the original manuscript, we did not mention the SSO. Now we added the description of SSO at the end of the third paragraph in introduction section: The most characteristic type of saltwater intrusion in the estuary is the saltwater-spill-over (SSO) from the North Branch into the South Branch (Shen et al., 2003; Wu et al., 2006; Lyu and Zhu, 2018). The shallow and funnel-shaped topography help to prevent runoff from entering the North Branch, especially during the dry season, and makes the tidal range larger in the North Branch than in the South Branch. The saltwater from the SSO is transported downstream by runoff and arrives in the middle reaches of the South Branch during the subsequent neap tide and threatens the water supplies of reservoirs located in the estuary.

**Comment 2:** Section 2.1: With regard to this section, I would suggest to move the analysis of the observed data into "Results" section 3.

**Reply:** Thanks. We moved the Section 2.1 into "Results" section 3.1 in the revised manuscript. The relevant section number was changed.

**Comment 3:** Calibration and Validation of the numerical model: Although the saltwater intrusion model has been extensively calibrated and validated, I would suggest the authors to present the calibration for the salinity along the channel (i.e., Baozhen, Nanmen, Chongxi and Qingcaosha). For the time being, we can only see the reproduction of the salinity in Baozhen station, while the performances in other available stations are unknown. In addition, it would be better to clarify the basic calibration parameters used in the numerical model in the manuscript.

**Reply:** It's a very good suggestion. In order to save the length of the original manuscript, we only presented the model validation for the salinity at Baozhen station. We first calibrated the model for water level and salinity by adjusting the basic parameters of bottom drag coefficient, vertical turbulent viscosity and diffusion coefficient used in the numerical model, then validated the model. Now we added the model validation for salinity at Baozhen, Nanmen, Chongxi and Qingcaosha stations

long the channel in Figure 3, and for water level at Baozhen, Sheshan and Luchaogang stations in Figure 6.

Because the model validation for salinity and water level at the hydrologic stations were added, the figures in the original manuscript were renumbered and redrew. Other figures are listed at the end of the reply.

[Figure]

Figure 3: Temporal variations in salinity in February 2014 at hydrologic stations. a: Baozhen station; b: Nanmen station; c: Chongxi station; c: Qingcaosha station. Black line: measured salinity; blue line: simulated salinity under climatic wind and residual water level conditions at open sea boundaries (Exp 1); red line: simulated salinity under a realistic wind and residual water levels at the open sea boundaries (Exp 2). The dashed green line represents salinity of 0.45, which is the standard for drinking water.

[Figure]

Figure 6: Temporal variations in water level in February 2014 at hydrologic stations. a: Baozhen station; b: Sheshan station; c: Luchaogang station. Black line: measured data; blue line: simulated data by Exp 1; red line: simulated data by Exp 2.

**Comment 4:** Section 4: In principle, the discussion part should systematically discuss the potential mechanism of saltwater intrusion with regard to the persistent and strong northerly wind. How the wind affects the individual terms in the momentum equation? The relationship between residual water level rise and wind? The generation of Ekman transport and the resulted horizontal circulation etc. In current form, the discussion part is more or less the same as the conclusions part since we see an identical sentence "With more frequent persistent and strong northerly wind caused by climate change, more attention should be paid to extremely severe saltwater intrusion events and freshwater safety in the Changjiang Estuary because the Qingcaosha Reservoir takes water from the Changjiang Estuary for the 13 million people in Shanghai" in both sections.

**Reply:** Thanks for your good suggestion. Now we added some numerical experiments and analyses in the discussion section to further reveal the potential mechanism of extremely severe saltwater intrusion in the revised manuscript. The

discussion 4 was rewritten as follows.

**4 Discussion**

In this section, the potential mechanism of the extremely severe saltwater intrusion with regard to the SSO, how the wind affects the individual terms in the momentum equations, relationship between residual water level rise and wind, generation of Ekman transport and the resulted horizontal circulation, and how long the northerly wind will induce severe saltwater intrusion were systematically discussed.

**4.1 How much contribution the SSO had in the saltwater intrusion event?**

The most feature of saltwater intrusion in the Changjiang Estuary is the SSO. Previous studies showed that the SSO is the main source of saltwater intrusion in the upper and middle reaches of the South Branch and the main saltwater source of the reservoirs (Shen et al. 2003; Wu et al. 2006; Zhu et al., 2013; Lyu and Zhu, 2018). How much contribution the SSO had in the extremely severe saltwater intrusion in February 2014?  A transect Sec 2 at the upper reaches of the North Branch (location labelled in Fig. 1) was set to calculate water and salt flux.

In Exp 1, the water flux from February 6 to 8, 2014 and salt flux on February 7, 2014 was transported from the South Branch into the North Branch. This phenomenon was occurred in neap tide (indicated by the water level in Fig. 6), while in the other tidal patterns the water and salt flux was transported from the North Branch into the South Branch, especially during the spring tide from February 14 to 17, 2014. This is the SSO occurring during middle and spring tide in dry season.

In Exp 2, the seaward water flux on February 7 became smaller and the landward water flux became larger from February 4 to 6 and from February 8 to 14, and the salt flux from February 4 to 14 was landward, under strong northerly wind, which was distinctly larger than the result under climatic wind and residual water level conditions at open sea boundaries. Therefore, the strong northerly wind enhanced the SSO, but was not the cause of the extremely severe saltwater intrusion event.

[Figure]

Figure 10: Temporal variations in residual water flux (a) and salt flux (b) across Sec 2 at the North Branch in February 2014. Dashed line: Exp 1; solid line: Exp 2. A positive value represents seaward flux, and a negative value represents landward flux.

A numerical experiment was designed in which the upper reaches of the North Branch is blocked (location labelled in Fig. 1) to further distinguish the contribution of SSO in the saltwater intrusion event. The distribution of time-averaged surface salinity from February 10 to 13, 2014 (Fig. 11a) and the temporal variations in salinity from February 8 to 22, 2014 at Chongxin station (Fig. 11b) show that the SSO was completely disappeared, while the salinity in the river mouth and near the Qingcaosha reservoir, and at the Baozhen, Nanmen and Qingcaosh stations were almost same as the results with Exp 2 (Fig. 7d, Fig. 3), indicating that the SSO almost had no contribution in the saltwater intrusion event in February 2014.

[Figure]

Figure 11: Distribution of time-averaged surface salinity from February 10 to 13, 2014 (a); and temporal variations in salinity from February 8 to 22, 2014 at hydrologic stations (b) if the upper reaches of the North Branch is blocked, and under a realistic wind and residual water levels at the open sea boundaries. Black line: Baozhen; red line: Nanmen; green line: Chongxi; blue line: Qingcaosha.

**4.2 How the wind affects the individual terms in the momentum equations?**

The modelled individual terms in the momentum equations were output at site Mo in the North Channel (labelled in Fig. 1, water depth 9.62 m) to analyse how the wind affects them. The momentum along the river channel ($\xi$ direction) is:

[revised manuscript text omitted]

In Exp 2, the situation in Exp 1 was changed greatly (Fig. 12c, d). The persistent and strong wind induced larger southward wind stress (red solid line in Fig. 12d), produced landward Ekman transport and water level rise, which significantly reduced the seaward barotropic pressure gradient force, especially from February 7 to 8, 2014. At this time, the barotropic pressure gradient force was very small, even negative (landward), that made the net water transport landward, greatly enhancing the saltwater intrusion in the North Channel, meanwhile distinctly increasing the landward baroclinic pressure gradient force (black dashed line in Fig. 12c).

[Figure]

Figure 12: Temporal variations in the vertically and time averaged terms in the momentum equations along the channel (left panel, a and c) and across the channel (right panel, b and d) from February 5 to 13, 2014 in Exp 1 (upper panel) and Exp 2 (lower panel) at site Mo in the North Channel.   Red solid line: wind stress at river surface; red dashed line: bottom friction force; black solid line: barotropic pressure gradient force; black dashed line: baroclinic pressure gradient force; green solid line: Coriolis force; green dashed line: advection; blue solid line: local variation; blue dashed line: horizontal turbulence viscosity.

**4.3 The relationship between residual water level rise and wind**

The water level rise in the Changjiang Estuary is closely related with the wind. The water level at Sheshan and Luchaogang station rose during northerly wind and dropped during southerly wind in February 2014 (Fig. 4b, c and d). High water level rise of more than 50 cm corresponded to the strong northerly wind. The water level is approximately 20 cm (Fig. 5b) under climatic wind in February and 30 cm (Fig. 9b) under persistent and strong northerly wind in February 2014. At Baozhen, Sheshan and Luchaogang station in February 2014, the modelled water level under climatic wind is lower than the one under strong northerly wind, especially from February 6 to 10, 2014, the difference reached 10~30 cm (Fig. 6). Therefore, the stronger the northerly wind is, the higher the water level in the Changjiang estuary and adjacent sea rises.

**4.4 The generation of Ekman transport and the resulted horizontal circulation**

The northerly wind produces southward current along the China coast, and push

water landward transport by Coriolis force, resulting in water level rise along the coast. In the Changjiang Estuary, this dynamic mechanism causes a pure wind-driven horizontal circulation that flows into the North Channel and out of the South Channel (Fig. 13a). The wind-driven estuarine current can enhance saltwater intrusion in the North Channel, and weaken it in the South Channel. Previous studies revealed the dynamic mechanism of northerly wind on the saltwater intrusion by the pure wind-driven current in the estuary (Wu et al., 2010; Li et al, 2012). In this study, the horizontal estuarine circulation was a total (net) circulation induced by the river discharge, tide and persistent and strong northerly wind (Fig. 9c), was not a pure wind-driven circulation, which surpassed the strong seaward runoff from February 9 to 13, 2014 (Fig. 8a). This result was unexpected and surprising, and was found for the first time.

A numerical experiment was set up with different northerly wind speed to ascertain how much wind can make the water flux at sec1 in the North Channel landward transport. Except for the wind, the other dynamic factors are the same as in Exp 2. The results is that the net water flux is seaward when wind is less than 8 m/s, is approximately zero when wind speed is 10 m/s, and landward when wind speed is larger than 10 m/s (Fig. 13b). The landward water flux enhances with increase of northerly wind. Therefore, northerly wind with speed of more than 10 m/s can called strong wind that causes net water transport landward in the North Channel.

[revised manuscript text omitted]

---

## Referee Comment (RC4) · Anonymous Referee #3 · 15 Aug 2020

**Comments to authors' reply**

For research, the most important thing is honesty and credit. In previous comments, I have pointed out several unreliable presentations. In authors' reply, there is still unreliable presentations such as the following reply to major issue 2. So, I cannot be sure if the modelling process was proper and the data used was correct. In addition, about some authors' explanation, I disagree. Authors did not seriously modify the manuscript. Therefore, I think the manuscript is not proper for publication. My main specific comments are as follows.

1. Reply to major issue 1

   What authors should do is adding introduce about previous related research, instead of only adding several references.

2. Reply to major issue 2

   Authors added the wind speed curve at weather station in figure R3-2 and said that the data used was 2 minutes average. I also have the wind observations data. At this land-based station inside estuary, it is impossible that the 2 minutes average wind speeds are so large, persistent more than 10 m/s for long time and even larger than 15 m/s. In figure R3-1, why did authors present the modeled wind directions and speeds off the Subei coast, instead of off the Changjiang Estuary?

3. Reply to major issue 4

   About why the water level rise inside estuary is small, authors said that I misunderstood and water level rise at Sheshan and Luchaogang stations in Figure 2d was almost same as the one in

Figure 5a. Authors clearly said in the manuscript that the water level rise at Sheshan and Luchaogang are distinct with a peak value more than 0.5 m (line 91), which is shown in plot d of figure 2 as well. But at Baozhen the relatively large rise during neap tide 7-11 is about 0.15 m (line 135), which can be seen from plot a of figure 5 as well. So, what is the real situation?

About the method used calculating water level rise in plot d of figure 2, authors said that they subtracted the data in the tide table from the measured water level value. The obtained water level rises based on this method could have much error because the forecasted water levels in tide table have error as well.

About much more water level rise inside the estuary in plot b of Figure 4 and Figure 7, authors said that in plot b of Figure 4 and plot b of Figure 7, the time-averaged water level was shown, not the water level rise. But it can be seen clearly that "water level rise" was labeled in the legend.

4. Reply to major issue 6

About already presented in previous work and unmentioned mechanism proposed in this manuscript, authors argued that the previous work was pure wind-driven and the work in this manuscript was not only wind-driven. But the proposed mechanism is the same. There is no the new thing, such as interaction between wind, tide, and river discharge. If you thought they were different, why did you not mention the previous work? Even if they are the same, it is ok only if you introduce and discuss the work. But you did not do this.

Authors said that if the wind directions were not always northerly, even southerly in some periods, the saltwater intrusion would be

more severe and more serious impact on the Qingcaosha reservoir. Why? If it is true, the mechanism should be shown as well.

In addition, it can be seen clearly from figure R3-4 that at Chongxi station the increase of salinity relative to the normal situation on 3-4 was more than the "extreme event" period. On 3-4 the strong northerly or northeasterly winds occurred as well. Why was the saltwater intrusion in the North Branch during the extreme event period is not extremely serious? During 13-17, salinity was similar to the normal situation, which means that there is no increase of saltwater intrusion. However, saltwater intrusion at other stations did not occur on 3-4, but was very serious on the "extreme event" period. What is the difference between mechanisms of winds influencing the North Branch and the North Channel?

---

## Author Response (AR1)

**Author's Response to Reviewers**

**Reply to anonymous referee #1**

Dear referee,

We appreciate your valuable questions and comments on our manuscript. The manuscript has been carefully revised according to your advice, and the questions and comments are marked with red (with blue and green for the other two reviewers). We believe your suggestions have improved our manuscript on the underlying mechanism of extremely severe saltwater intrusions and the role of the persistent and strong northerly wind. The point-by-point answers to the questions and comments are listed as follows.

Sincerely,

Jianrong Zhu, on behalves of the co-authors

**Comment 1:** Section 2.1: It can be seen from observed salinity (in Figure 2a) that at normal condition the salinity at the upstream Chongxi station is higher than other stations, while during the abnormal condition (i.e., the studied case) the maximum salinity concentration appears at Baozhen station. I would suggest the authors to clarify the difference between these two conditions since the underlying mechanism for saltwater intrusion may be completely different. For instance, to what extent the spilling of saltwater from the North Branch into the South Branch may impact the dynamics of saltwater concentration during this abnormal condition when

compared with the normal condition.

**Reply:** Thanks for your good comment. Under normal conditions with climatic wind in the dry season, there exists saltwater spillover (SSO) from the North Branch into the South Branch during spring tide, which causes the salinity at Chongxi station to exceed the levels at Nanmen and Baozhen stations. During the abnormal condition with a persistent and strong northerly wind in February 2014, the maximum salinity was considerably higher at Baozhen station than at Chongxi station. To quantitatively analyze of impact of the SSO on the saltwater intrusion in the South Branch, the temporal variations in residual water flux and salt flux across sec2 (newly labeled in Figure 1) at the upper reaches of the North Branch in February 2014 were modeled under climatic wind and real wind conditions in February 2014. In addition, a numerical experiment in which the upper reaches of the North Branch is blocked was designed to determine the effect of the SSO on saltwater intrusion in the South Branch under the real wind in February 2014. The above contents have been added to the discussion section 4.1 in the revised manuscript.

In the original manuscript, we did not mention SSO. Now, we added the description of SSO at the end of the third paragraph in the introduction section on lines 69-75: "The most characteristic type of saltwater intrusion in the estuary is the saltwater spillover (SSO) from the North Branch into the South Branch (Shen et al., 2003; Wu et al., 2006; Lyu and Zhu, 2018). The shallow and funnel-shaped topography helps to prevent runoff from entering the North Branch, especially during the dry season, and produces a greater the tidal range in the North Branch than in the

South Branch. The saltwater from the SSO is transported downstream by runoff and arrives in the middle reaches of the South Branch during the subsequent neap tide, threatening the water supplies of reservoirs located in the estuary."

**Comment 2:** Section 2.1: With regard to this section, I would suggest to move the analysis of the observed data into "Results" section 3.

**Reply:** Thanks. We moved Section 2.1 into results section 3.1 in the revised manuscript. The relevant section number was changed.

**Comment 3:** Calibration and Validation of the numerical model: Although the saltwater intrusion model has been extensively calibrated and validated, I would suggest the authors to present the calibration for the salinity along the channel (i.e., Baozhen, Nanmen, Chongxi and Qingcaosha). For the time being, we can only see the reproduction of the salinity in Baozhen station, while the performances in other available stations are unknown. In addition, it would be better to clarify the basic calibration parameters used in the numerical model in the manuscript.

**Reply:** This is a very good suggestion. To preserve the length of the original manuscript, we only presented the model validation for the salinity at Baozhen station. We first calibrated the model for water level and salinity by adjusting the basic parameters of bottom drag coefficient, vertical turbulent viscosity and diffusion coefficient used in the numerical model, then validated the model. Now, we added the model validation for salinity at Baozhen, Nanmen, Chongxi and Qingcaosha stations

along the channel in Figure 3 and for water level at Baozhen, Sheshan and Luchaogang stations in Figure 6 in the revised manuscript, and presented the results of model validation on lines 227-236 in the revised manuscript.

Because the model validations for salinity and water level at the hydrologic stations were added, the figures in the original manuscript were renumbered and redrawn.

**Comment 4:** Section 4: In principle, the discussion part should systematically discuss the potential mechanism of saltwater intrusion with regard to the persistent and strong northerly wind. How the wind affects the individual terms in the momentum equation? The relationship between residual water level rise and wind? The generation of Ekman transport and the resulted horizontal circulation etc. In current form, the discussion part is more or less the same as the conclusions part since we see an identical sentence "With more frequent persistent and strong northerly wind caused by climate change, more attention should be paid to extremely severe saltwater intrusion events and freshwater safety in the Changjiang Estuary because the Qingcaosha Reservoir takes water from the Changjiang Estuary for the 13 million people in Shanghai" in both sections.

**Reply:** Thanks for your good suggestion. We added the following content to discuss the potential mechanism of saltwater intrusion with regard to the persistent and strong northerly wind in the discussion section of the revised manuscript:

1) To quantitatively analyze the impact of the SSO on the saltwater intrusion in

the South Branch, the temporal variations in residual (net) water flux and salt flux across sec2 at the upper reaches of the North Branch in February 2014 under climatic wind and real wind in February 2014 were added, and the analyses are presented in Section 4.1 in the revised manuscript.

In addition, we added a numerical experiment in which the upper reaches of the North Branch are blocked to distinguish the effect of the SSO on saltwater intrusion in the South Branch under the real wind in February 2014. The results are shown in Figure 12.

2) The individual terms in the momentum equations were output to discuss the wind effect on the saltwater intrusion.

This content was added in Section 4.2. Temporal variations in the vertically and time averaged terms in the momentum equations along the channel and across the channel from February 5 to 13, 2014, under climatic wind (Exp 1) and real wind (Exp 2) at a site in the North Channel are shown in Figure 13. How the wind affected the terms in the momentum equations is discussed.

3) Relationship between residual water level and wind

The residual (mean) water level in the Changjiang Estuary is closely correlated with the wind. The water level at Sheshan and Luchaogang stations rose during the northerly wind and dropped during the southerly wind in February 2014 (Fig. 4d). The water level rise of more than 50 cm corresponded to the strong northerly wind. The water level was approximately 20 cm (Fig. 5b) under climatic wind in February and 30 cm (Fig. 9b) under persistent and strong northerly wind in February 2014. At

Baozhen, Sheshan and Luchaogang stations in February 2014, the modeled water level under climatic wind was lower than that under strong northerly wind, especially from February 6 to 10, 2014, when the difference reached 10-30 cm (Fig. 6). Therefore, the stronger the northerly wind is, the higher the water level in the Changjiang Estuary and adjacent sea.

The above part was added in the revised manuscript in discussion section 4.3.

4) The generation of Ekman transport and the resulting horizontal circulation

This suggestion was added in discussion section 4.4. The distribution of pure wind-driven current time averaged from February 10 to 13, 2014, modeled under the real wind, and the relationship between water flux across sec1 in the North Channel and the northerly wind speed are presented in Figure 14.

5) The sentence "With more frequent persistent and strong northerly wind caused by climate change, more attention should be paid to extremely severe saltwater intrusion events and freshwater safety in the Changjiang Estuary because the Qingcaosha Reservoir takes water from the Changjiang Estuary for the 13 million people in Shanghai" in the discussion section was removed the revised manuscript.

The contents added in the discussion section are summarized in the conclusion section and presented in the abstract.

**Reply to anonymous referee #2**

Dear referee,

We appreciate your valuable questions and comments on our manuscript. The manuscript has been carefully revised according to your advice, and the revisions are marked with blue (with red and green for the other two reviewers). The point-by-point answers to the questions and comments are listed as follows.

Your sincerely,

Jianrong Zhu, on behalves of the co-authors

**Comment 1:** According to Figure 5b, the realistic wind can be a cause. However, what is the role of the spring tide (The period from 10 Feb to 18 Feb 2014 according to the lunar calendar) for the intrusion during this issue?

**Reply:** As we can see from Figure 5b, the strongest saltwater intrusion occurred from February 11 to 14, 2014, in the middle tide, corresponding to lunar calendar dates 12 to 15. The tidal pattern of the middle tide is indicated by the measured tidal level at Baozhen hydrologic station in Figure 5a. The highest tidal level is on lunar calendar date 18, not 15, in the Changjiang Estuary. There are four tidal patterns in one neap-spring tide: neap tide, middle tide after neap tide, spring tide, and middle tide after spring tide. Each tidal pattern lasts approximately 4 days. Our study indicated that the saltwater intrusion event was caused by the persistent and strong northerly wind, which played a more important role in the saltwater intrusion event during neap tide than spring tide.

**Comment 2:** Had some issues also happened in the near Hangzhou Bay? Since

the wind also causes a rise of residual water level there (Figures 4, 7, 8). If yes please add some details; if no what caused the different issues between the Changjiang Estuary (saltwater intrusion) and Hangzhou Bay (no issue) since the wind would be equal for these two.

**Reply:** Thanks for your good comment. Hangzhou Bay is adjacent to the Changjiang Estuary and is occupied by seawater all year round. The salinity is considerably higher than 0.45 (the standard salinity for drinking water), and there is no freshwater in Hangzhou Bay. Therefore, there is no study of saltwater intrusion there. It is true that the strong northerly wind also caused a rise in residual water level and an increased salinity in Hangzhou Bay.

The Qiantang River connects Hangzhou Bay, and saltwater intrusions often occur in the dry season. The most downstream water plant, Nanxing water plant, is located in the Qiantang River to supply water to Hangzhou (Figure R2-1, just for reply to referee, not included in the manuscript). Because the water intake of the water plant is far away from the river mouth and the river depth is only 1-2 m, the severe saltwater intrusion in the Changjiang Estuary in February 2014 did not occur there. The saltwater intrusion was not as severe in the North Branch of the Changjiang Estuary because the water depth is very shallow there, as is the case for the Qiantang River. In the North Branch of the Changjiang Estuary, the landward wind-driven Ekman water transport was weaker, flowing along the north side and flowing out along the south side only near the river mouth (Fig. 7c in the revised manuscript). However, the North Channel is deeper and wider and located on the north side of the

South Branch, which is conducive to strong landward Ekman water transport in the North Channel.

[Figure]

Figure R2-1 Satellite image of the Changjiang Estuary, Changjiang Bay and Qiantang River

**Minor issue**

1, Line 239, Page 14, Check the name.

**Reply: Thanks.** The name "Lijinag" has been changed to "Linjiang."

**Reply to anonymous referee #3**

Dear referee,

We appreciate your valuable questions and comments on our manuscript. The manuscript has been carefully revised according to your advice, and the revisions are marked with green (with red and blue for the other two reviewers). Thank you again for your time in reviewing this manuscript. The point-by-point answers to the questions and comments are listed as follows.

Your sincerely,

Jianrong Zhu, on behalves of the co-authors

**First review**

**Major issues:**

**Major issue 1:** There are many studies about the influence of winds on saltwater intrusion in the estuary. But authors only mentioned two papers (Xue et al., 2009; Li et al., 2012) about the Changjiang Estuary. Even about the Changjiang Estuary, there are not only two papers.

**Reply:** Thanks for your good suggestion. We have cited the following papers about the influence of winds on saltwater intrusion in the estuary:

Aristizabal, M. F., and Chant, R. J. (2015): An observational study of salt fluxes in Delaware Bay, Journal of Geophysical Research-Oceans, 120, 2751-2768, 2015.

Chen, S. N., and Sanford, L. P.: Axial wind effects on stratification and longitudinal salt transport in an idealized, partially mixed estuary, Journal of Physical Oceanography, 39(8), 1905-1920, 2009.

Duran-Matute, M., Gerkema, T., and Sassi, M. G.: Quantifying the residual volume transport through a multiple-inlet system in response to wind forcing: The case of the western Dutch Wadden Sea, Journal of Geophysical Research: Oceans, 12, 8888-8903, 2016.

Regarding the Changjiang Estuary, the following papers on the influence of winds on saltwater intrusion have been cited:

Wu H., Zhu J. R., and Choi B. H.: Links between saltwater intrusion and subtidal circulation in the Changjiang Estuary: a model-guided study, Cont. Shelf Res., 30, 1891-1905, 2010.

Zhang E, Gao S, Savenije H. H. G, Sic C, and Cao S: Saline water intrusion in relation to strong winds during winter cold outbreaks: North Branch of the Yangtze Estuary, Journal of Hydrology, 574: 1099-1109, 2019.

Ding L., Dou X. P., Gao X. Y., Jiao J., and Hu J.: Response of salinity intrusion to winds in the Yangtze Estuary, Proceedings of 5th International Conference on Hydraulic Engineering, CHE2017, 241-247, 2017.

**Major issue 2:** About the strong wind event, it should be defined such as wind speed and duration. Authors said that from February 5 to 14, 2014 a persistent and strong northerly wind occurred lasting ten days. And they presented that only a strong northerly wind lasting 8 days can produce a severe saltwater intrusion in the Changjiang Estuary. But seen from plot c of Figure 2, on 5-6 the wind directions were southerly and easterly, and the winds seemed not strong. On 12-14, the winds were

not strong as well. Authors should show what magnitude of wind event could induce severe saltwater intrusion. In plot c of Figure 2, the curve of wind speeds should be added. Thus the magnitude of winds can be seen clearly. What kind of data was used in plot c, instantaneous value, 2 minutes average, or maximum in a gust of wind? They should be presented clearly. In addition, the weather station locates inside the estuary near the mouth. The wind direction at this station may be different from the sea.

**Reply:** Thanks for your good question. There is really no defined strength of wind causing severe saltwater intrusions in the Changjiang Estuary. From the observed and simulated wind and results of numerical experiments on the effect of wind on saltwater intrusion in the estuary, wind with a speed greater than 10 m/s and lasting 8 days can be called a strong wind.

We checked the wind in plot c of Figure 2; the wind was northeasterly on February 3 and 4 and was southeasterly on 5 with speed of approximately 10 m/s. The mean wind speed on February 12-13 was approximately 10 m/s and was only 5 m/s on February 14. In fact, the measured wind at the weather station on the Chongming eastern shoal was only used to describe the phenomenon of persistent and strong wind and was not used to simulate the severe saltwater intrusion event. The wind used in the model was that simulated by the WRF, in which the wind speed adjacent to the sea near the Changjiang River mouth was stronger than that on the Chongming eastern shoal (Figure R3-1). Now, the sentence "a persistent and strong northerly wind from February 5 to 14, 2014, lasting ten days" was changed to "a persistent and strong

northerly wind from February 6 to 14, 2014, lasting nine days". Other relevant parts of the manuscript have also been revised.

In plot c of Figure 2, the curve of wind speeds has been added (Figure R3-2). This figure was enlarged to clearly illustrate the wind vector and wind.

The wind data used in plot c were 2 minute averages and are illustrated in the revised manuscript.

Yes, the weather station is located inside the estuary near the mouth, and the wind direction at this station was somewhat different from that over the sea (seen in Figure R3-1 and Figure R3-2). It needs to be emphasized again that the wind used in the model was that simulated by the WRF, in which the wind speed over the sea was stronger than that on the Chongming eastern shoal.

[Figure]

Figure R3-1: Temporal variations in wind vector (a) and wind speed (b) simulated by the WRF model off the Subei coast in February 2014.

[Figure]

Figure R3-2: Temporal variations in the measured river discharge at Datong station (a), wind vector (b) and wind speed (c) at WS, and water level rise obtained by subtracting the data in the tide table from the measured water level at Sheshan station (black line) and Luchaogang station (red line) (d) in February 2014 in February 2014.

**Major issue 3:** This severe saltwater intrusion event is strange. The peak salinity at Baozhen and Nanmen stations reached 20.1 and 12.4 respectively. But why did the salinity only reach 8.6 at Qingcaosha? The location of Qingcaosha is close to Baozhen, and downstream of Nanmen. It can be seen from plot a of Figure 2 that salinity at

Qingcaosha was much lower than Baozhen. In addition, salinity at Chongxi station is from the North Branch. Before 7 February salinity was high at Chongxi, very low even close to zero at other stations. But during the severe event salinity was very high at other stations, but low at Chongxi station. These need explanation.

**Reply:** This is a good question and is exactly what this manuscript intends to reveal. From the longitudinal view, the extreme saltwater intrusion in February 2014 came from the downstream sea; consequently, the peak salinity at Baozhen, Nanmen and Chongxi stations was 20.1, 12.4 and 4.0 during the event, respectively.

From the transverse perspective, although the location of Qingcaosha is close to Baozhen and downstream of Nanmen, the salinity only reached 8.6 at Qingcaosha and was much lower than that at Baozhen. This was because the saltwater was brought by the flood current and moved rightward under the effect of Coriolis force, causing the salinity to be higher on the north side than on the south side of the North Channel. The simulated salinity distribution showed the same phenomenon. In both of the normal condition and the abnormal condition (i.e., the studied case), this transverse difference of salinity was same.

Under normal conditions with climatic wind in the dry season, there exists saltwater spillover (SSO) from the North Branch into the South Branch in spring tide, causing the salinity at Chongxi station to be higher than at Nanmen and Baozhen stations. This is the most obvious feature of saltwater intrusion in the Changjiang estuary and is a well-known phenomenon that increases the salinity at Chongxi station compared to those at Nanmen and Baozhen stations. During the abnormal condition

with persistent and strong northerly wind in February 2014, the opposite occurred. This is the phenomenon our paper seeks to reveal.

**Major issue 4:** Authors presented that the water level rose distinctly at the coast during the event (Figure 2, line 91). Why did the water level inside the estuary not rise obviously (Figure 5, lines 133-135)? Authors said that the water level inside the river mouth was mainly determined by tide and river discharge. This needs explanation. Seen from plot b of Figure 4 and plot b of Figure 7, the water level rise inside the estuary was much larger than the coast. This is inconsistent with authors' expression and observations at Baozhen. The water level rises in figure 7 and figure 8 seem the same both for 10-13 February. In addition, about plot d in Figure 2, how was the water level rise obtained or how did authors calculate the water level rise? This should be presented clearly in methods section.

**Reply:** Thanks. The two different concepts of water level and water level rise need to be distinguished here. The water level rise at Sheshan and Luchaogang stations in Figure 2d was almost the same as the high water level along the coast which was induced by the strong northerly wind shown in Figure 5a. We note that the water level at Baozhen station inside the river mouth was mainly determined by tide and river discharge because the temporal variations in the observed and modeled water level were roughly same.

In plot b of Figure 4 and plot b of Figure 7, the time-averaged water level from February 10 to 13, 2014, is shown, not the water level rise. The mean water level was

much higher inside the estuary than off the coast due to the river discharge.

The residual water level from February 10 to 13, 2014, in Figure 7b and Figure 8a is the same and was duplicated. These two figures were redrawn, plot a of Figure 8 was deleted, and the residual surface current was moved to plot b of Figure 7 (Figure R3-3).

[Figure]

Figure R3-3: Distributions of the temporally averaged wind field from February 7 to 14, 2014, as simulated by the WRF model (a), and the time-averaged water level and surface current from February 10 to 13, 2014, as simulated by the model encompassing the Bohai Sea, Yellow Sea and East China Sea (b).

Regarding plot d in Figure 2, the water level was provided by the Shanghai Hydrology Administration, and the water level rise was obtained by subtracting the data in the tide table from the measured water level value. This information is

provided in the revised manuscript.

**Major issue 5:** The main work of this manuscript is modeling of salinity and water level during the severe event. But authors only presented the results at Baozhen station (Figure 5). The results at other stations should be shown as well. Figures 6-8 only present the time-averaged results on10-13 February.

**Reply:** This is a good suggestion. Now, we present the measured results of salinity at Baozhen, Nanmen, Chongxi and Qingcaosha stations and of water level at Baozhen, Sheshan and Luchaogang stations (Figure R3-4 and Figure R3-5). The model was validated for salinity and water level (as Reviewer 1 suggested). The following contents marked with green were added in the revised manuscript:

The difference between the observed water level and modeled water level under climatic wind can be considered the water level rise by the northerly strong wind. It is seen from the Figure R3-5 that the water level rise at low water level from February 5 to 12, 2014 was approximately 0.35, 0.40, and 0.30 m at Baizhen, Sheshan and Luchaogang station, respectively.

[Figure]

Figure R3-4: Temporal variations in salinity in February 2014 at hydrologic stations. a: Baozhen station; b: Nanmen station; c: Chongxi station; c: Qingcaosha station. Black line: measured salinity; blue line: simulated salinity under climatic wind and residual water level conditions at open sea boundaries; red line: simulated salinity under a realistic wind and residual water levels at the open sea boundaries. The dashed green line represents salinity of 0.45, which is the standard for drinking water.

[Figure]

Figure R3-5: Temporal variations in water level in February 2014 at hydrologic stations. a: Baozhen station; b: Sheshan station; c: Luchaogang station. Black line: measured data; blue line: simulated data under climatic wind and residual water level conditions at open sea boundaries; red line: simulated data under a realistic wind and residual water levels at the open sea boundaries.

**Major issue 6:** The dynamic mechanism of the severe saltwater intrusion event is the objective of this manuscript. The manuscript proposed the mechanism: landward Ekman transport forms a horizontal estuarine circulation that flowed into the North Channel and out of the South Channel. This mechanism or result is not new. It has been presented in authors' previous work (Wu Hui, Zhu Jianrong, Choi Byung Ho, 2010. Links between saltwater intrusion and subtidal circulation in the Changjiang Estuary: a model-guided study. Cont. Shelf Res. 30 (17): 1891–1905.).

But authors did not mention this work. This reference did not occur in the manuscript as well. The strong winds were not persistent for very long time. And the wind directions were not always northerly, even southerly in some periods. Why did the severe saltwater intrusion last 23 days? This is the question the manuscript should answer. About the results presented in discussion part and figure 9, it is doubtful. About plot b, can two-day strong winds induce the higher than normal salinity in after 8 days? About plots c and d, there are similar doubts. What is the mechanism of this? This needs detailed explanation. In addition, many studies mentioned that water withdrawal between Datong and estuary could increase saltwater intrusion in dry season. Is there possibility that during the severe event water withdrawal downstream Datong was large contributing to this event as well?

**Reply:** This is a misunderstanding. Wu et al. (2010) did present a horizontal estuarine circulation that flowed into the North Channel and out of the South Channel; rather, the circulation they described was a purely wind-driven current under a northerly wind of 7 m/s (Figure 10 in their paper). Li et al. (2012) also presented a pure wind-driven current (Figure 14 in their paper). This pure wind-driven current was used to explain how the northerly wind affected the saltwater intrusion in the Changjiang Estuary. The pure wind-driven estuarine current can enhance saltwater intrusion in the North Channel and weaken it in the South Channel. In this study, the horizontal estuarine circulation was a total (net) circulation induced by the river discharge, tide and persistent and strong northerly wind and was not a pure wind-driven circulation, which surpassed the strong seaward runoff. This result was

unexpected and was found for the first time. The relevant illustration was added in the revised manuscript. The paper of Wu et al. (2010) was cited in the revised manuscript.

It is rare for a northerly wind with speed of greater than 10 m/s and lasting more than 9 days to occur in the Changjiang Estuary, but it did occur (Figure R3-2) and caused the severe saltwater intrusion in February 2014, influencing the water intake of Qingcaosha reservoir for 23 days. If the wind directions were not always northerly, even southerly in some periods, the saltwater intrusion would be more severe and more serious impact on the Qingcaosha reservoir. This manuscript reveals why the extreme saltwater intrusion occurred as well as the dynamic mechanism.

In Figure 9, the strong northerly wind lasts two, four, six and eight days in plot a, b, c and d, respectively. Two-day strong winds are in plot a, not in plot b. It is true that several days of strong wind can induce the higher than normal salinity after 8 days because the strong wind Ekman transport brought the salinity front in a sandbar area upstream and closer to the Baozhen station (model output site in Figure 9) and then moved upstream and downstream with the oscillation of flood and ebb currents for 8 days when the northerly wind is 5 m/s. It will take approximately 8 days to eliminate the impact of the upstream forward salinity front on the salinity at Baozhen station. This phenomenon was also shown in Figure 2a, 2c. The relevant illustration was added in the revised manuscript.

Yes, many studies pointed out that water withdrawal between Datong and the estuary could increase saltwater intrusion in dry season, but there was no possibility that the water withdrawal downstream Datong had a large contribution to the severe

saltwater intrusion event because the water withdrawal amount by the South-to-North Water Diversion Project and water diversion and drainage along the river was approximately 1000 m³/s, and the monthly mean river discharge was 12,430 m³/s.

**Major issue 7:** The structure of the manuscript is strange. In section 2.1 observed data, authors introduced the severe saltwater intrusion events, which should be moved to introduction section or results section. In section 3 results, section 3.1 is not necessary. The results under normal situations and special situations can be compared in order to show the difference. But the result under normal situation is not the important results for the objective of the manuscript. In other words, it is not necessary presented separately. In addition, the discussion part is too simple.

**Reply:** Thanks for your good suggestion. We moved Section 2.1 on observed data to section 3 of the results in the revised manuscript and deleted the headings of 3.1 Climatic wind and residual water level conditions at open sea boundaries and 3.2 Under a realistic wind in February 2014 and residual water levels at the open sea boundaries in the revised manuscript, and add the follow contents as the first paragraph of the section 3.2 Model results:

To reveal which dynamic factor caused the extremely severe saltwater intrusion event in February 2014, two numerical experiments were designed. The first experiment considered climatic wind conditions at the sea surface and residual water levels at open sea boundaries (Exp 1), and the second considered a realistic wind in February 2014 and residual water levels at the open sea boundaries (Exp 2). The river

discharge at the upper boundary at Datong station and tide at the open sea boundary were the same in Exp 1 and Exp 2.

According to Reviewer 1 and your suggestions, the discussion part has been greatly expanded. The contribution of the SSO to the saltwater intrusion event, how the wind affects the individual terms in the momentum equations, the relationship between residual water level and wind, and the generation of Ekman transport and the resulting horizontal circulation were added and discussed in the revised manuscript.

**Major issue 8:** Some presentations or data in the manuscript are unreliable. Besides some mentioned above examples are as follows.

(1) About the data source, in section 2.1 (observed data), authors said that the observed data was conducted by State Key Laboratory of Estuarine and Coastal Research, East China Normal University. But in acknowledgements part authors said that the observed data was provided by Shanghai Hydrology Administration.

(2) Page 3, line 57, "... caused by very low river discharge of approximately 7000 and 8000 $m^3s^{-1}$ lasting three mouths (mouths should be months), respectively". In this sentence, "8000 $m^3s^{-1}$ in 1999" and "lasting three months" are not correct. In dry season of 1979, river discharges in January and February were really very low between 7000 and 8000 $m^3s^{-1}$ at Datong station. In March the monthly mean discharge was more than 10000 $m^3s^{-1}$ during which severe saltwater intrusion also occurred. In 1999 the monthly mean river discharges at Datong were all larger than 9000 $m^3s^{-1}$. In February extremely severe saltwater intrusion occurred as well

inducing continuous 25 days of unsuitable drinking water at Chenhang Reservoir upstream of Qingcaosha, during which discharge was 9110 $m^3s^{-1}$.

**Reply:** Thanks for pointing out the incorrect expressions in the original manuscript.

(1) In Section 2.1 (observed data), we modified the expression as follows: the observed salinity and wind data were from the State Key Laboratory of Estuarine and Coastal Research, East China Normal University, and the observed water level data were from the Shanghai Hydrology Administration. In the acknowledgements part, we modified the expression as follows: the authors thank the Shanghai Hydrology Administration for providing the observed water level in the Changjiang Estuary.

(2) We checked and drew the variation processes of river discharge measured at Datong station in dry seasons of 1979 and 1999 (Figure R3-6). In the dry season of 1979 from January 1 to March 15, the river discharge at Datong station was between 6000 and 9280 $m^3s^{-1}$ with mean value of 7485 $m^3s^{-1}$, lasting 2.5 months. In the dry season of 1999 from January 1 to March 16, the river discharge at Datong station was between 7650 and 1,0900 $m^3s^{-1}$ with mean value of 9246 $m^3s^{-1}$, lasting 2.5 months.

Therefore, we modified the sentence "Two severe saltwater intrusion events were recorded in the Changjiang Estuary in the dry seasons of 1979 and 1999, which were caused by very low river discharge of approximately 7000 and 8000 $m^3s^{-1}$ lasting three months" to "Two severe saltwater intrusion events were recorded in the Changjiang Estuary in the dry seasons of 1979 and 1999, which were caused by very low river discharge with a mean value of 7485 $m^3 s^{-1}$ from January 1 to March 15,

1979, and 9246 m$^3$s$^{-1}$ from January 1 to March 16, 1999, both lasting 2.5 months".

The word "mouths" has been changed to "months."

[Figure]

Figure R3-6: Temporal variations in river discharge measured at Datong station from December 1, 1978 to April 30, 1979 (a), and from December 1, 1998 to April 30, 1999 (b).

**Minor issues:**

**Minor issue 1:** In page 2, lines 30-31, what is the maximum spring tide and minimum neap tide? The proper expressions should be the maximum tidal range of spring tide and minimum tidal range of neap tide. Or, the maximum tidal range and minimum tidal range are enough.

**Reply:** Thanks. "the maximum spring tide" was changed to "the maximum tidal range", and "the minimum neap tide" was changed to "minimum tidal range".

**Minor issue 2:** In page 3, lines 49-50, "which was the largest estuarine reservoir in the world, the Qingcaosha Reservoir, was built". There is syntax error in this sentence.

**Reply:** "which was the largest estuarine reservoir in the world, the Qingcaosha Reservoir, was built" was changed to "when the largest estuarine reservoir in the world, the Qingcaosha Reservoir, was built".

**Minor issue 3:** In the last paragraph of introduction, authors said that an extremely severe saltwater intrusion event in February 2014 occurred, and this is a catastrophic event never occurred. But we did not see how severe and catastrophic. The compare between this event and historical severe events should be presented as well in order to show the severe magnitude.

**Reply:** The extremely severe saltwater intrusion event in February 2014 resulted in the continuous period of unsuitable drinking water reaching 23 days and caused a serious threat to the water intake of the Qingcaosha Reservoir and water safety in Shanghai. Therefore, we refer to the saltwater intrusion as a catastrophic event. Historically, there were severe saltwater intrusions, such as in dry seasons of 1979 and 1999, but there was no reservoir in the Changjiang Estuary at that time. Thus, we only called these saltwater intrusions severe events. A catastrophic event refers to water security.

**Minor issue 4:** About the river discharge used during modelling, did you consider the time required for water traveling from Datong to the estuary? Usually the discharges several days in advance are used because Datong station is located more than 600 km upstream of the estuary.

**Reply:** This is a good question. The west open boundary of the Changjiang River in the model is up to Datong station in the Changjiang River. Thus, it is not necessary to consider the time required for water traveling from Datong to the estuary. The sentence "Datong hydrological station is on the western open boundary of the Changjiang River" was added at the end of the first paragraph in section 2.1 numerical model.

**Minor issue 5:** Caption of Figure 2: Temporal variations in the measured data in February. Plot d does not present the measured data. The water level rises are calculated results.

**Reply:** Thanks. Now the caption of Figure 2 was changed to "Temporal variations in the measured river discharge at Datong station (a), wind vector (b) and wind speed (c) at WS, and water level rise obtained by subtracting the data in the tide table from the measured water level at Sheshan station (black line) and Luchaogang station (red line) (d) in February 2014. The figure number is changed to Figure 4 in the revised manuscript.

**Minor issue 6:** Caption of Figure 3 is not clear. And there are syntax errors.

**Reply:** The caption of Figure 3 was rewritten. The Figure 3 is changed to Figure 2 in the revised manuscript.

Figure 2 Model grids of the Changjiang Estuary (a), and model grids of the Bohai Sea, Yellow Sea and East China Sea (b). Domains of the models (c); within the red line: the Changjiang Estuary model domain; within the green line: model domain of the Bohai Sea, Yellow Sea and East China Sea; the black dashed lines: the two-fold nested WRF model domain.

**Minor issue 7:** What is the climatic wind? This expression is strange.

**Reply:** Climate wind means monthly mean wind for many years.

**Minor issue 8:** Captions of Figures 5, 6, 8 are not clear.

**Reply:** The captions of Figures 5, 6, 8 were rewritten. The figures were reorganized and redrawn, and the modified captions were presented in the revised manuscript.

**Second review**

**Referee's reply to major issue 1:** What authors should do is adding introduce about previous related research, instead of only adding several references.

**Author's reply:** Thanks your comment. We didn't make it clear about citing more references of influence of winds on saltwater intrusion in the estuary. We thought the revised manuscript will be uploaded and the referee will see the detailed

modification. We cited the references and briefly described them in the introduce section besides in the inferences section.

The sentences in the original manuscript are rewritten to the following sentences in introduce section in the revised manuscript (the green parts are the added contents): Saltwater intrusion is a common phenomenon in estuaries where fresh water and saltwater converge, and is mainly controlled by tide and river discharge (Prandle, 1985; Simpson et al., 1990; Geyer, 1993), but it can also be affected by wind stress (Chen and Sanford, 2009; Aristizábal and Chant, 2015; Duran-Matute et al., 2016; Giddings and Maccready, 2017) and vertical mixing (Simpson and Hunter, 1974; Prandle and Lane, 2015).   Along-estuary winds can strain density gradients, and the associated destruction or enhancement of stratification depending on wind direction, and the entrainment depth ratio (Chen & Sanford, 2009). Wind-driven sea-level setups at the mouth of estuaries can produce landward flows that outcompete river runoff, resulting in the net, landward advection of salt (Aristizabal & Chant, 2015). For multi-inlet coastal systems, residual (horizontal) circulation is influenced by winds, which alters salt transports in each inlet (Duran-Matute et al., 2016). The upwelling and downwelling favorable wind can significantly influence the estuarine exchange flow (Giddings, and MacCready, 2017).

For the Changjiang Estuary, the sentences in the original manuscript are modified to the following sentences in introduce section in the revised manuscript (the green parts are the added contents): Saltwater intrusion in the Changjiang Estuary is also mainly determined by river discharge and tides (Song and Mao, 2002; Gu et al,

2003; Shen et al., 2003; Luo and Chen, 2005; Qiu et al. 2012; Chen et al, 2019a) but

is also influenced by wind (Xue et al., 2009; Wu et al., 2010; Li et al., 2012; Ding et

al., 2017; Zhang et al., 2019), and topography (Li et al., 2014; Chen et al., 2019b).

The impact of wind on saltwater intrusion has been studied, but only with a climatic

wind (Xue et al., 2009; Qiu et al. 2012; Chen et al, 2019a ), and a strong northerly

wind induced by ordinary cold fronts in winter lasting 1-2 days, which could cause a

change in the observed salinity (Li et al., 2012). Xue et al. (2009) pointed out that a

northerly wind tends to enhance the saltwater intrusion in the North Branch by

reducing the seaward surface elevation gradient forcing. Wu et al. (2010) and Li et al.

(2012) simulated the pure wind-driven current that flows into the North Channel and

out of the South Channel with climatic wind to explain that the northerly wind can

enhance saltwater intrusion in the North Channel, and weaken it in the South Channel.

Zhang et al. (2019) reported that the frequency of saltwater intrusion events in the

Changjiang Estuary is increasing in recent years due to increasing frequency of winter

storms passing East China Sea.

**Referee's reply to major issue 2:** Authors added the wind speed curve at

weather station in figure R3-2 and said that the data used was 2 minutes average. I

also have the wind observations data. At this land-based station inside estuary, it is

impossible that the 2 minutes average wind speeds are so large, persistent more than

10 m/s for long time and even larger than 15 m/s. In figure R3-1, why did authors

present the modeled wind directions and speeds off the Subei coast, instead of off the

Changjiang Estuary?

**Author's reply:** We established and managed the weather station at the Chongming eastern shoal for more than 15 years. The wind direction and speed was recorded with several forms, i.e., instantaneous, 2 minutes average, 10 minutes average, and maximum. We used the 2 minutes averaged wind direction and speed. The monthly mean wind in the Changjiang river mouth is 5.5 m/s during winter and 5.0 m/s in summer. February 2014 is a special month, occurred persistent and strong northerly wind. In Fact, the weather station is just on the river mouth (shown in Figure 1), not inside the estuary. The reviewer said he also has the wind observations data. Would you please draw a figure of the wind vector and speed curve to see how weak your wind is? And where is the weather station? This is a major issue the reviewer proposed, we will very appreciate you if the question can be figured out and what is the really wind.

We presented the figure R3-1 only to indicate the wind was much stronger on the sea than at Chongming eastern shoal for the reviewer. This picture is not used in the manuscript, only appears in the author's reply. We did not instead of the observed wind at the weather station with the modeled wind on the seas. We want to emphasize again that the observed wind data is only used to illustrate the wind status, and the wind used in the saltwater intrusion model was the simulated wind field by the WRF. The model domain is large, and wind has spatial variation. A point wind cannot represent a wind field. It can been seen from the figure 7a that the temporally averaged wind field from February 7 to 14, 2014 simulated by the WRF reached more

than 15 m/s in the Yellow Sea and off the Changjiang river mouth, indicating that there did exist a long persistent and strong northerly wind in February, 2014.

**Referee's reply to major issue 3:** About why the water level rise inside estuary is small, authors said that I misunderstood and water level rise at Sheshan and Luchaogang stations in Figure 2d was almost same as the one in 2 Figure 5a. Authors clearly said in the manuscript that the water level rise at Sheshan and Luchaogang are distinct with a peak value more than 0.5 m (line 91), which is shown in plot d of figure 2 as well. But at Baozhen the relatively large rise during neap tide 7-11 is about 0.15 m (line 135), which can be seen from plot a of figure 5 as well. So, what is the real situation?

About the method used calculating water level rise in plot d of figure 2, authors said that they subtracted the data in the tide table from the measured water level value. The obtained water level rises based on this method could have much error because the forecasted water levels in tide table have error as well.

About much more water level rise inside the estuary in plot b of Figure 4 and Figure 7, authors said that in plot b of Figure 4 and plot b of Figure 7, the time-averaged water level was shown, not the water level rise. But it can be seen clearly that "water level rise" was labeled in the legend.

**Author's reply:** Thanks again. We checked the water level rise in Figure 2d and water level in in Figure 5a, and concluded that the water level rise at Sheshan and Luchaogang are distinct with a peak value more than 0.5 m (Figure 2d). At Baozhen

station, the difference between the observed water level and modeled water level under climatic wind can be roughly considered the water level rise by the northerly strong wind. It is seen from the Fig. 5a that the water level rise at low water level from February 5 to 12, 2014 was approximately 0.35 m.

We agree the reviewer's opinion that the forecasted water levels in tide table have error. Because the water level can be directly measured, the water level rise cannot be obtained by direct observation, and the forecasted water levels in tide table without strong northerly wind considered, the water level rise is roughly reasonable with the method subtracted the data in the tide table from the measured water level value.

Thank you pointing out the mistake. The legend in Figure 4b and in Figure 7b should be mean water level. Now we modified it in the revised manuscript.

**Referee's reply to major issue 4:** About already presented in previous work and unmentioned mechanism proposed in this manuscript, authors argued that the previous work was pure wind-driven and the work in this manuscript was not only wind-driven. But the proposed mechanism is the same. There is no the new thing, such as interaction between wind, tide, and river discharge. If you thought they were different, why did you not mention the previous work? Even if they are the same, it is ok only if you introduce and discuss the work. But you did not do this.

Authors said that if the wind directions were not always northerly, even southerly in some periods, the saltwater intrusion would be more severe and more serious

impact on the Qingcaosha reservoir. Why? If it is true, the mechanism should be shown as well.

In addition, it can be seen clearly from figure R3-4 that at Chongxi station the increase of salinity relative to the normal situation on 3-4 was more than the "extreme event" period. On 3-4 the strong northerly or northeasterly winds occurred as well. Why was the saltwater intrusion in the North Branch during the extreme event period is not extremely serious? During 13-17, salinity was similar to the normal situation, which means that there is no increase of saltwater intrusion. However, saltwater intrusion at other stations did not occur on 3-4, but was very serious on the "extreme event" period. What is the difference between mechanisms of winds influencing the North Branch and the North Channel?

**Author's reply:** Thanks your comment. Now we added the previous work of the wind impact on the saltwater intrusion in the Changjiang Estuary in introduction section. Xue et al. (2009) pointed out that a northerly wind tends to enhance the saltwater intrusion in the North Branch by reducing the seaward surface elevation gradient forcing. Wu et al. (2010) and Li et al. (2012) simulated the pure wind-driven current that flows into the North Channel and out of the South Channel with climatic wind to explain that the northerly wind can enhance saltwater intrusion in the North Channel, and weaken it in the South Channel. In the discussion section in the revised manuscript, we added the following sentences marked with green:

The wind-driven estuarine current can enhance saltwater intrusion in the North Channel, and weaken it in the South Channel. Previous studies revealed the dynamic

mechanism of northerly wind on the saltwater intrusion by the pure wind-driven current in the estuary (Wu et al., 2010; Li et al, 2012). In this study, the horizontal estuarine circulation was a total (net) circulation forced by the river discharge, tide and persistent and strong northerly wind (Fig. 8a), which surpassed the strong seaward runoff.

I'm sorry we didn't make it clear of the impact of wind direction on saltwater intrusion. The correct expression is: If the wind directions were always northerly, and the southerly in some periods is northerly, the saltwater intrusion would be more severe and more serious impact on the Qingcaosha reservoir.

Thank you asking a very good question of the different performance of saltwater intrusion between at Chongxi station and at Nanmen, Baozhen station. The saltwater intrusion at Chongxi station is completely from the North Branch (SSO, saltwater-spill-over from the North Branch into the South Branch), while at Namen and Baozhen station is mainly from the SSO under normal wind condition. Because the North Branch is very shallow, the landward wind-driven Ekman water transport in it was weaker, flowing along the north side and flowing out along the south side only near the river mouth (Fig. 8b), the persistent and strong wind cloud enhance the SSO, but not so much significant as in the North Channel. The North Channel is deeper and wider and located on the north side of the South Branch, which is conducive to strong landward Ekman water transport in the North Channel, resulting in severe saltwater intrusion. Therefore, the saltwater intrusion in the North Branch during the extreme event period is not extremely serious.

At Chongxi station the increase of salinity relative to the normal situation on February 3-4 was more than the "extreme event" period, this is because the northerly wind was strong which has somewhat influence on the saltwater intrusion, but the major cause was the asymmetry of the semi lunar spring tide. This can be confirmed in Figure 5a that the high water level was higher on February 3-4 than on February 18-19 in the spring tide.

In order to better reveal the contribution of the SSO in the saltwater intrusion event in February 2014, we added contents in the discussion section 4.1 in the revised manucript. Reviewer 1 asked the same quiestion, so the contents are marked with red in the revised manuscript. The contents are as follows.

4.1 What was the contribution of the SSO to the saltwater intrusion event?

The most obvious feature of saltwater intrusion in the Changjiang Estuary is the SSO. Previous studies showed that the SSO is the main source of saltwater intrusion in the upper and middle reaches of the South Branch and the main saltwater source of the reservoirs (Shen et al. 2003; Wu et al. 2006; Zhu et al., 2013; Lyu and Zhu, 2018). What was the extent of the contribution of the SSO to the extremely severe saltwater intrusion in February 2014? A transect Sec 2 at the upper reaches of the North Branch (location labeled in Fig. 1) was set to calculate water and salt flux.

In Exp 1 (under climatic wind and residual water level conditions at open sea boundaries), the water flux from February 6 to 8, 2014, and salt flux on February 7, 2014, were transported from the South Branch into the North Branch (Figure R3-7). This phenomenon occurred during a neap tide (indicated by the water level in Figure

R3-5), while in the other tidal patterns, the water and salt flux was transported from the North Branch into the South Branch, especially during the spring tide from February 14 to 17, 2014. This is the famous SSO occurring during middle and spring tide in dry season.

In Exp 2 (under a realistic wind and residual water levels at the open sea boundaries), the seaward water flux on February 7 decreased and the landward water flux increased from February 4 to 6 and from February 8 to 14, and the salt flux from February 4 to 14 was landward, under strong northerly wind, which was distinctly greater than the result under climatic wind and residual water level conditions at open sea boundaries. Therefore, the strong northerly wind enhanced the SSO.

[Figure]

Figure R3-7: Temporal variations in residual water flux (a) and salt flux (b) across Sec 2 in the North Branch in February 2014. Dashed line: Exp 1; solid line: Exp 2. A positive value represents seaward flux, and a negative value represents landward flux.

A numerical experiment was designed in which the upper reaches of the North Branch were blocked (location labeled in Fig. 1 in the revised manuscript) to further distinguish the contribution of SSO in the saltwater intrusion event. The distribution of time-averaged surface salinity from February 10 to 13, 2014 (Fig. R3-8a) and the temporal variations in salinity from February 8 to 22, 2014 at Chongxin station (Fig. R3-8b) show that the SSO was completely absent, while the salinity in the river mouth and near the Qingcaosha reservoir and at the Baozhen, Nanmen and Qingcaosh stations was almost identical to the results from Exp 2 (Fig. 8c, Fig. R3-4), indicating that the SSO had almost no contribution to the saltwater intrusion event in February 2014.

[Figure]

Figure R3-8: Distribution of time-averaged surface salinity from February 10 to 13, 2014 (a); and temporal variations in salinity from February 8 to 22, 2014 at hydrologic stations (b) if the upper reaches of the North Branch is blocked, and under a realistic wind and residual water levels at the open sea boundaries. Black line: Baozhen; red line: Nanmen; green line: Chongxi; blue line: Qingcaosha.

**Third review**

**Comments to authors' reply 1:** The following figure is the wind observations (2 minutes average) in February 2014. I think the location is the same as yours, located at Chongming eastern shoal as well. The shape of wind speed curve is similar to yours, but the magnitude is much weaker than yours. I also have observations at other stations along the coast, with similar magnitude even weaker. Even if the observed wind data is only used to illustrate the wind status, they should be true. In addition, why did you delete one day data (wind and water level rise on 1) in figure R3-2? But the date still began from 1. It seems that the data or figure can be changed as you want.

[Figure]

**Reply:** Thank you presenting the wind observations at Chongming eastern shoal in February 2014. It showed that the mean wind speed was approximately 7.5 m/s in February 2014 and weaker than our observed one. The wind simulated by the WRF off the Subei coast (the site was labeled in Figure R3-9) in February 2014 was shown in Figure R3-1, indicating that the wind in the Yellow Sea near the North Branch was much stronger than the observed one at Chongming eastern shoal. Now we download the wind from the European Centre for Medium-Range Weather Forecasts (ECMWF, 2014; http://apps.ecmwf.int/datasets) and choice the point which is closest the Sheshan island (labeled in Figure R3-9) to present the temporal variations in wind vector and wind speed in February 2014 (Figure R3-10). It can be seen that the wind is stronger than the observed one provided by the reviewer, and is little weaker than the author's observed one. The data of ECMWF was a post processed data and its quality is widely accepted. There existed some dissimilar among the winds from observed, simulated and ECMWF, but the phenomenon of persistent and strong wind is same. We quite agree with the reviewer's opinion that even if the observed wind data is only used to illustrate the wind status, they should be true. The following sentence was added on lines 223-226 in the revised manuscript: The temporal variations in wind vector and wind speed (b) at Sheshan island (labeled in Fig. 1) in February 2014 simulated by WRF (Fig. 10) was similar with the one observed at Chongming eastern shoal (Fig. 4b, c), indicating again that there existed a persistent and strong northerly wind in February 2014.

[Figure]

Figure R3-9: Locations of the weather station at Chongming eastern shoal, WRF

output site of WRF (the triangle on the north side) and output site of ECMWF.

[Figure]

Figure R3-10: Temporal variations in wind vector (a) and speed (b) download from ECMWF in February 2014.

Thank you for pointing out the problem. The data in Figure R3-2 was from February 2, and now the data on February 1 was added.

**Comments to authors' reply 2**:About the modeled wind directions and speeds (figure R3-1), even if they are only shown for referee, the location or area should be off Changjiang estuary instead of Subei coast. I think it is necessary shown in the manuscript in order to show the winds you used in the model. It can be seen from figure R3-1 that the wind directions were almost all northerly. Some were not only different from station located at Chongming eastern shoal but also different from the station outside the estuary. For example, on 5-6 winds are not strong with directions of southeasterly and easterly at station near the mouth, easterly outside the estuary, but winds are strong with directions of northerly in figure R3-1 (modeled winds). On 10-11, the wind directions are northwesterly at station outside the estuary, but northerly in figure R3-1. The wind directions and speeds observed at station outside the estuary could induce water level setdown, which is consistent with the calculated water level change (I ever did). It can be seen also from plot d of figure 2 that on 5-6 and 10-11 the water level did not rise. This means the modeled wind directions and speeds may be not correct in some periods, which will induce incorrect results.

**Reply:** Thank you for your comment. Now we presented the figure of temporal variation in wind vector and speed modeled by WRF at Sheshan island that is just off the Changjiang Estuary (Figure 10 in the revised manuscript).

Yes, the modeled wind off the Subei coast (Figure R3-1) had some difference with the observed one at Chongming eastern shoal, indicating that the wind had spatial variation. That is why we used the WRF to simulate the wind to better reproduce spatial and temporal variation in wind. The water level change (plot d of figure 2) was certainly affected by the wind. We agree the reviewer's opinion that if the modeled wind directions and speeds are not correct in some periods, it will induce incorrect results for the model results. We have been using the WRF model for more than 15 years and can certain that this model is reliable. The wind simulated by WRF at Sheshan island in February 2014 (Figure 10) was very consistent with the one near the Sheshan island download from ECMWF (Figure R3-10). The salinity variation processes at the hydrologic stations in February 2014 (Figure 3 in the revised manuscript) were successfully repeated, indicating that the wind simulated by WRF was believable because the severe saltwater intrusion was caused by the wind.

**Comments to authors' reply 3:** About the water level rise shown in plot b of figures 4 and 7, authors said it is the mistake, and the "water level rise' in legend should be "mean water level". But the maximum value in legend is 0.5 m, which should not be mean water level.

**Reply:** Because there existed tide in the total water level, the tidal fluctuation was filtered out in in plot b of figures 4 and 7, the water level was a residual or mean water level. The maximum value in legend is 0.5 m, meaning that the mean water level near the coast is higher, that was caused by the northerly wind. In the caption of plot b of figures 7, there is "the time-averaged water level and surface current from February 10 to 13, 2014", meaning it was a mean water level. Now the number of figure 4 and figure 7 in the original manuscript was changed to figure 5 and figure 9 in the revised manuscript, and the modified captions were marked with green.

**Comments to authors' reply 4:** About error of calculated water level rise in plot d of figure 2, the method used is the oldest method, authors should try other method. Authors' some argument about water level rise is not consistent with the text.

**Reply:** Because the water level rise cannot be directly observed, and the observed one is the total water level. We agree that method of water level rise by subtracting the data in the tide table from the measured water level is an old method, but it can roughly reflect the water level rise. The water level rise can be obtained with numerical simulation by subtracting the mean water level in Figure 5b from the mean water level in Figure 9b (in revised manuscript), which can be roughly considered the water level rise caused by the strong northerly wind. If the wind is set to 0 m/s in Figure 5b, then difference is the exact water level rise induced by the strong northerly wind.

**Comments to authors' reply 5:** About the mechanism of the extreme event, I think it was still not clear. Authors said that the strong northerly winds lasting 4 days can induce the higher than normal salinity in after 8 days (plot b of figure 9). In plot b the winds were set to 5 m/s beginning 9 February. But the strong northerly winds began from 7 before which the wind directions were southerly or easterly both at station near the mouth and station outside the estuary. This means that the real strong northerly winds lasted 2 days. This is why I ask the question (About plot b, can two-day strong winds induce the higher than normal salinity in after 8 days?). Authors replied that the two-day strong winds are in plot a, not in plot b. Now I know the reason, the modeled strong northerly winds lasted 4 days before 9.

**Reply:** The mechanism of the extreme event is that it was caused by the persistent and strong northerly wind, which drove substantial landward net water transport to form a horizontal estuarine circulation that flowed into the North Channel and out of the South Channel. This landward net water transport overpowered the seaward-flowing river runoff and transported a large volume of highly saline water into the North Channel. Figure 9 was used to discuss how long the northerly wind can induce severe saltwater intrusions. We check again the wind in Figure R3-10 and in Figure 10 in the revised manuscript, and sure that the northerly wind began on February 6. So, the strong northerly wind in Figure 9a, b, c, and d lasted 2, 4, 6 and 8 days, respectively.

**Comments to authors' reply 6**: About the North Branch, even if the impact of winds on saltwater intrusion is weaker than the North Channel, the saltwater intrusion should be also much stronger than normal situation during the"extreme event" period. It can be seen from figure R3-4 the salinity on 13-16 at Chongxi station was similar to the normal situation, but the salinity at other stations dramatically increased.

**Reply:** Yes, the saltwater intrusion in the North Branch was enhanced by the strong northerly wind in February, 2014. In Figure R3-4, the salinity on February 13-16 at Chongxi station was similar to the normal situation, but the salinity at other stations dramatically increased, just meaning that the SSO is somewhat enhanced, but the saltwater intrusion was extremely severe at other stations because the strong northerly wind produced a net horizontal circulation that 
[revised manuscript text omitted]